# One-Sided Matrix Completion from Ultra-Sparse Samples

**Hongyang R. Zhang**                                                       *ho.zhang@northeastern.edu*
*Northeastern University, Boston*

**Zhenshuo Zhang**                                                       *zhang.zhens@northeastern.edu*
*Northeastern University, Boston*

**Huy L. Nguyen**                                                       *hu.nguyen@northeastern.edu*
*Northeastern University, Boston*

**Guanghui Lan**                                                       *george.lan@isye.gatech.edu*
*Georgia Institute of Technology, Atlanta*

**Reviewed on OpenReview:** *https://openreview.net/forum?id=vYGi4Dj777*

## Abstract

Matrix completion is a classical problem that has received recurring interest from a wide range of fields. In this paper, we revisit this problem in an *ultra-sparse sampling regime*, where each entry of an unknown, $n \times d$ matrix $M$ (with $n \geq d$) is observed independently with probability $p = C/d$, for a fixed integer $C \geq 2$. This setting is motivated by applications involving large, sparse panel datasets, where the number of rows (users) far exceeds the number of columns (items). When each row contains only $C$—fewer than the rank of $M$—accurate imputation of $M$ is impossible. Instead, we focus on estimating the *row span* of $M$, or equivalently, the averaged *second-moment matrix* $T = M^\top M/n$.

The empirical second-moment matrix computed from observational data exhibits non-random and sparse missingness. We propose an *unbiased estimator* that normalizes each nonzero entry of the second moment by its observed frequency, followed by gradient descent to impute the missing entries of $T$. This normalization divides a weighted sum of $n$ binomial random variables by their total number of ones—a nonlinear operation. We show that the estimator is unbiased for any value of $p$ and enjoys low variance. When the row vectors of $M$ are drawn uniformly from a rank-$r$ factor model satisfying an incoherence condition, we prove that if $n \geq O(dr^5\epsilon^{-2}C^{-2}\log d)$, any local minimum of the gradient-descent objective is approximately global and recovers $T$ with error at most $\epsilon^2$.

Experiments on both synthetic and real-world data validate our approach. On three Movie-Lens datasets, our algorithm reduces bias by 88% relative to several baseline estimators. We also empirically evaluate the linear sampling complexity of $n$ relative to $d$ using synthetic data. Finally, on the Amazon reviews dataset with sparsity $10^{-7}$, our method reduces the recovery error of $T$ by 59% and $M$ by 38% compared to existing matrix completion methods.

## 1 Introduction

Matrix completion is a classical problem that has received sustained attention across several disciplines, including compressive sensing (Candes & Plan, 2010), machine learning (Srebro et al., 2004), and signal processing (Chi et al., 2019). A seminal result in the literature states that when the unknown matrix $M$ is low rank and satisfies an incoherence assumption (i.e., the row norms of the low-rank factors are approximately balanced), exact recovery from a uniformly random subset of entries is achievable (Candes & Recht, 2012).

In this paper, we study matrix completion in an *ultra-sparse sampling regime*, where the sampling probability $p = C/d$, for some constant $C \geq 2$, resulting in roughly $Cn$ observed entries in total. This is less than

$nr \log d$, the number of bits required to store the rank-$r$ factors of an $n$ by $d$ matrix. Although recovering $M$ itself is information-theoretically impossible (Chen, 2015), we show that it is possible to recover **one side** of $M$—specifically, the second-moment matrix $T = M^\top M / n$—in this regime. In other words, when $M$ is a tall matrix, our goal is to recover the *subspace spanned by the row vectors* of $M$.

There are several motivations for revisiting matrix completion in the ultra-sparse sampling regime. First, many large-scale panel datasets are extremely sparse in practice. For example, consider the Amazon reviews dataset (Hou et al., 2024), which contains roughly 54 million users and 48 million items, with about 571 million observed reviews in total. The resulting sparsity level is as low as $10^{-7}$. Therefore, we need matrix completion methods that can remain effective under this level of ultra-sparsity. Second, many recommendation systems must preserve data privacy. In such settings, each user may contribute only a small portion of data to a central server, often after adding white noise for differential privacy (Liu et al., 2015). The central server then aggregates these contributions by computing summary statistic such as the *low-rank subspace spanned by the row vectors.* This summary statistic can then be shared publicly, allowing each user to project their data locally onto the learned subspace (Wang et al., 2023). In these scenarios, the number of users can be orders of magnitude larger than the number of items (see Table 2 for concrete examples), which motivates studying recovery of the row space of $M$ when $n \gg d$.

Recent work by Cao et al. (2023) initiates the study of *one-sided matrix completion*, whose goal is to estimate the averaged second-moment matrix of the row vectors of $M$, denoted by

$$T = \frac{1}{n} M^\top M,$$

under the setting where two entries per row are observed. In other words, the matrix $T$ captures the row-space structure of $M$. Their analysis leaves open the general case where more than two entries are observed per row. To illustrate the challenge, let $\widehat{M}$ denote the observed data matrix and $I$ the corresponding binary mask indicating observed entries. Because $\widehat{M}$ is highly sparse, the information it contains is insufficient to recover $M$ directly—rendering standard matrix completion guarantees inapplicable to this regime. Let $\Omega$ denote the set of nonzero entries of the empirical second-moment matrix $\widehat{M}^\top \widehat{M}$, which has the same dimension as $T$. We make two key observations about $\widehat{M}^\top \widehat{M}$ that together highlight the technical challenges in recovering $T$ under ultra-sparse sampling:

1. The missing patterns in $\Omega$ is highly non-random: diagonal entries are almost always observed, while off-diagonal entries are extremely sparse. For a detailed calculation, see Claim 2.1. Moreover, the observation frequencies in the off-diagonal entries of $I^\top I$ are sparse and unevenly distributed.

2. Although stochastic gradient algorithms are widely used in practice (Lan, 2020), providing theoretical guarantees for matrix completion is technically challenging (Sun & Luo, 2016) and has not been done under non-random missingness, to the best of our knowledge.

To address the first issue, we analyze a weighted estimator that normalizes each nonzero entry of $\hat{M}^\top \hat{M}$ by the corresponding entry at $I^\top I$. This approach corresponds to the Hájek estimator (Hájek, 1971; Särndal et al., 2003), which divides two random quantities—making its analysis more subtle due to nonlinearity. *Our key observation is that the Hájek estimator is exactly unbiased on the observed subset $\Omega$, unlike prior results that show only asymptotical unbiasedness as $n \to \infty$.* This finite-sample unbiasedness arises from the symmetry of pairwise products in computing the second-moment matrix.

Furthermore, a first-order approximation analysis on the variance of the Hájek estimator reveals that this estimator achieves *lower variance* than the classical Horvitz-Thompson estimator (Horvitz & Thompson, 1952), leading to more stable and less biased empirical estimates, a result corroborated by our experiments.

To tackle the second issue, we establish *recovery guarantees for any local minimizer of a loss function defined between the normalized empirical second-moment matrix and its low-rank factorization* used to impute missing entries of $T$ that lie outside $\Omega$. Under a rank-$r$ factor model for the rows of $M$, we show that if

$$n \geq O\big(dr^5 C^{-2} \epsilon^{-2} \log d\big),$$

Table 1: Comparison between this work and several closely related results on matrix completion under an ultra-sparse sampling regime. Here, $M \in \mathbb{R}n \times d$ with $n \geq d$, and $T = \frac{1}{n} M^\top M$ denotes the averaged second-moment matrix of the rows of $M$. We focus on the one-sided or ultra-sparse settings where exact recovery of $M$ itself is infeasible.

| Estimand | Sampling Regime | Main Approach (Reference) |
|:---:|:---:|:---:|
| $M$ | $p = O(\frac{1}{n})$ | Thresholded Alternating Minimization (Gamarnik et al., 2017) |
| $T$ | Two entries per row | Nuclear-Norm Regularization (Cao et al., 2023) |
| $T$ | $p = \frac{C}{n}, \forall C \geq 2$ | HÁJEK-GD with Incoherence Regularization (This paper) |

then any local minimizer of the loss objective is within $\epsilon^2$ Frobenius norm distance to $T$. The big-$O$ notation above ignores dependencies of the bound on the condition number of $M$ as well as the incoherence condition number (See Assumption 3.3 for the definition). This bound assumes standard incoherence conditions—without which recovery is information-theoretically impossible (Candes & Recht, 2012)—and is optimal in scaling relative to $d$. Importantly, the result holds for any constant $C$, thus generalizing prior results and closing an open question left by Cao et al. (2023). A key technical step is establishing a concentration inequality to tackle the non-random missingness of $\Omega$, by leveraging a conditional independence property at each row of $\widehat{M}^\top \widehat{M}$. This conditional independence property enables us to analyze and bound the bias of the Hájek estimator row by row, ensuring consistency even under heterogeneous observation patterns.

**Numerical results.** We extensively validate our approach on both synthetic and real-world datasets. We find that the Hájek estimator is particularly effective under extreme sparsity. For instance, when each row has only two observed entries, our method outperforms a prior nuclear-norm regularization baseline by 70% in recovering $T$. We also explore using one-sided matrix completion as a subroutine for imputing missing entries of $M$. On the Amazon reviews dataset, we extend our estimator to impute $M$ via least-squares regression after estimating $T$, achieving 21% and 38% lower root mean squared error (RMSE) than alternating gradient descent and soft-impute with alternating least squares (Hastie et al., 2015; Chi et al., 2019; Li et al., 2020a), respectively. Ablation studies further show that the Hájek estimator reduces bias on $\Omega$ by 99% relative to the Horvitz-Thompson estimator on synthetic data, and by 88% on average across three MovieLens datasets. Incorporating an incoherence regularization term (Ge et al., 2016) into the loss further decreases recovery error of $T$ by 22% on synthetic data and by 52% on the Amazon review dataset.

**Summary of contributions.** This paper revisits the matrix completion problem under an ultra-sparse sampling regime, where only $Cn$ entries are observed from an unknown $n \times d$ matrix for a constant $C \geq 2$. Our contributions are threefold:

1. We design a new algorithm for one-sided matrix completion based on the Hájek estimator, accompanied by a detailed theoretical analysis. We prove that the Hájek estimator is an unbiased estimator of the second-moment matrix for any sampling probability $p$ and achieves lower variance than the Horvitz-Thompson estimator.

2. We establish near-optimal sample complexity bounds for one-sided matrix completion under a low-rank, incoherent factor model. Our results apply to gradient-based optimization and extend to general sampling patterns beyond those considered in prior work. See Table 1 for a comparison of sample complexity and main approach.

3. We perform extensive experiments on synthetic and real-world datasets, demonstrating the practical effectiveness of our approach and showing that one-sided matrix completion can be used as a robust subroutine for full matrix recovery. The implementation and reproducible code are available at: `https://github.com/VirtuosoResearch/Matrix-completion-ultrasparse-implementation`.

One limitation of our sample complexity result lies in the common-means assumption in the rank-$r$ factor model. We believe this condition can be relaxed if each factor is perturbed by random noise (see Remark 3.5). Another limitation concerns the existence of exact low-rank factor models for recovering $T$ and the need

to specify the true rank in the gradient-descent procedure (see Remark 4.3). We discuss possible extensions to address these issues in Section 7.

**Organizations.** The remainder of this paper is organized as follows. Section 2 describes the problem setup and notations. Section 3 presents the estimator and the main theoretical results. Section 4 outlines the proof sketch and two extensions. Section 5 and 6 provide empirical results and related work, respectively. Section 7 concludes the paper and discusses several open directions. Complete proofs appear in Appendix A-B, and additional experiments are discussed in Appendix C.

## 2 Preliminaries and Notations

Let $M$ denote an $n$ by $d$, unknown matrix. Assume the rank of $M$ is equal to $r$. Without loss of generality, suppose that $n \geq d$. For example, each row of $M$ may include the ratings of a user, while each column may correspond to the ratings of an item. We observe a partial matrix, denoted by $\widehat{M}$. Let $I \in \{0, 1\}^{n \times d}$ denote the indicator matrix on the observed entries of $\widehat{M}$. Our goal is to recover the *averaged second-moment matrix* $T := M^\top M / n$, given $\widehat{M}$.

We assume that each entry of $\widehat{M}$ is observed independently with probability $p \in (0, 1)$, following prior literature (e.g., Candes & Recht (2012); Sun & Luo (2016); Ge et al. (2016)). We focus on a constant sampling regime, where $p = C/d$, for some fixed integer $C \geq 2$. In such a regime, $m$ is roughly $Cn$, less than the number of bits to represent an $n$ by $r$ matrix.

An important quantity for working with one-sided estimation is $I^\top I$, which is a symmetric matrix in dimension $d$. This matrix provides the empirical frequency counts for the empirical second moment matrix $\widehat{M}^\top \widehat{M}$. For example, consider an off-diagonal entry $(i, j)$, where $i \neq j$ and both $i, j$ are in $\{1, 2, \ldots, d\}$. The $(i, j)$-th entry of $I^\top I$, denoted as $(I^\top I)_{i,j}$, is equal to the number of overlapping, nonzero entries between the $i$-th and $j$-th column of $\widehat{M}$. That is,

$$\left(I^\top I\right)_{i,j} = \sum_{k=1}^{n} I_{k,i} I_{k,j}. \tag{1}$$

Let $\Omega$ denote the indices of the nonzero entries of $I^\top I$. We now make the following observation regarding the observed entries within $\Omega$. In particular, we find that the sampling patterns of $\Omega$ are non-random in the following sense.

**Claim 2.1.** *For every $i = 1, 2, \ldots, d$, the diagonal entries satisfy that*

$$\Pr[(i, i) \in \Omega] = 1 - (1 - p)^n.$$

*The off-diagonal entries satisfy that for every $i = 1, \ldots, d$, and $j = 1, \ldots, d$ such that $j \neq i$,*

$$\Pr[(i, j) \in \Omega] = 1 - (1 - p^2)^n.$$

To illustrate, recall that $p = C/d$. Suppose $n$ is at the order of $O(d \log d)$. The probability that $(i, i) \in \Omega$ is roughly $1 - e^{-pn} \approx 1 - d^{-O(C)}$. Then by a union bound, with probability at least $1 - O(d^{-1})$, it must be the case that $(i, i) \in \Omega$, for every $i = 1, 2, \ldots, d$.

For each off-diagonal entry, it is observed in $\Omega$ with probability equal to $1 - (1 - p^2)^n$. For simplicity of notation, let us denote this as $q$ in the rest of the paper. Note that $q = np^2$ is roughly equal to $O(C^2 d^{-1} \log d)$.

To account for the difference in sampling probabilities between diagonal and off-diagonal entries of $\Omega$, let $P_\Omega : \mathbb{R}^{d \times d} \to \mathbb{R}^{d \times d}$ be a weighted projection operator defined as follows:

$$(P_\Omega(Z))_{i,j} = \begin{cases} q Z_{i,i}, & \text{if } (i, i) \in \Omega, \\ Z_{i,j}, & \text{if } (i, j) \in \Omega \text{ and } j \neq i, \\ 0, & \text{if } (i, j) \notin \Omega. \end{cases}$$

**Estimators.**  Having introduced the co-occurrence matrix $I^\top I$ and its nonzero index set $\Omega$, next, we discuss the normalization of the empirical second moment of the observed matrix, given by $\widehat{M}^\top \widehat{M}$. One way to normalize $\widehat{M}^\top \widehat{M}$ for one-sided matrix completion is via the Horvitz-Thompson (HT) estimator (Horvitz & Thompson, 1952), which inversely reweights each entry with the true sampling probability (i.e., $p$ on the diagonal entries and $p^2$ on the off-diagonal entries):

$$\overline{T}_{i,j} = \begin{cases} \dfrac{\sum_{k=1}^n M_{k,i}^2 I_{k,i}}{np}, & \text{if } i = j, \\[2ex] \dfrac{\sum_{k=1}^n M_{k,i} M_{k,j} I_{k,i} I_{k,j}}{np^2}, & \text{if } i \neq j. \end{cases}$$

One can verify that $\mathbb{E}[\overline{T}_{i,j}] = T_{i,j}$. Notice that $np^2$ is the expectation of $(I^\top I)_{i,j}$ (cf. equation (1)), while $np$ is the expectation when $i = j$. When $p$ is too small, $np^2$ becomes small, resulting in high fluctuation in the off-diagonal entries. Another approach, known as the Hájek estimator (Hájek, 1971), is to normalize each entry in $\Omega$ by the estimated sampling probability, which is canceled out since the probability value is the same across different $k$, leading to the following expression, for any $(i, j) \in \Omega$:

$$\widehat{T}_{i,j} = \begin{cases} \dfrac{\sum_{k=1}^n M_{k,i}^2 I_{k,i}}{\sum_{k=1}^n I_{k,i}}, & \text{if } i = j, \\[2ex] \dfrac{\sum_{k=1}^n M_{k,i} M_{k,j} I_{k,i} I_{k,j}}{\sum_{k=1}^n I_{k,i} I_{k,j}}, & \text{if } i \neq j. \end{cases}$$

The theoretical analysis of the Hájek estimator is challenging due to its nonlinear form, which involves dividing one random variable by another. It is known that the Hájek estimator is asymptotically unbiased (Hájek, 1971) in the large $n$ limit, incurring an error of $O(n^{-1/2})$. In addition, the Hájek estimator provides a variance reduction effect compared to the Horvitz-Thompson estimator in causal inference (Hirano et al., 2003). Little is known in the matrix completion problem, and several natural questions arise. First, how does this estimator work for one-sided matrix completion? What is the bias of $\widehat{T}$, and how does the Hájek estimator compare to the Horvitz-Thompson estimator? Second, how can we use $\widehat{T}$ to impute the missing entries outside $\Omega$? The rest of this paper is dedicated to answering these questions.

**Notations.**  Before continuing, we provide a list of notations for describing the results. Following the convention of big-$O$ notations, given two functions $f(n)$ and $g(n)$, let $f(n) \leq O(g(n))$ indicate that there exists a constant $c$ independent of $n$ such that when $n \geq n_0$ for some large enough $n_0$, then $f(n) \leq c \cdot g(n)$. We use the notation $f(n) \lesssim g(n)$ as a shorthand for indicating that $f(n) = O(g(n))$. Let $\tilde{O}(g(n))$ denote that $O(\log^c(n)g(n))$ for some constant $c$ independent of $n$. We also use the little-o notation $f(n) \leq o(g(n))$, which means that $f(n)/g(n)$ goes to zero as $n$ goes to infinity.

Let $[d] = \{1, 2, \ldots, d\}$ denote a shorthand notation for the set from 1 up to $d$. We use $\mathcal{N}(a, b)$ to denote a Gaussian distribution with mean set at $a$ and variance equal to $b$. We use the notation $(x)_+ = \max(x, 0)$ to denote a truncation operation set above zero.

Let $\|\cdot\|_F$ denote the Frobenius norm of an input matrix. Let $\|\cdot\|_2$ denote the spectral norm of the input matrix and let $\|\cdot\|$ denote the Euclidean norm of a vector. Let $\langle X, Y \rangle = \text{Tr}[X^\top Y]$ denote the inner product between two matrices that have the same dimensions. Let $\|\cdot\|_\infty$ denote the infinity norm of a matrix, which corresponds to its largest entry in absolute value.

## 3  Estimation and Recovery Guarantees

In this section, we present a thorough analysis of the Hájek estimator in the one-sided matrix completion problem. We begin by analyzing the bias of $\widehat{T}$ compared to $T$ on the observed entries $\Omega$. A key observation is that $\widehat{T}$ provides an unbiased estimate of $T$, as stated formally below.

**Lemma 3.1.** *Suppose the entries of $\widehat{M}$ are sampled from $M$ independently with a fixed probability $p \in (0, 1)$. Let $\Omega$ denote the index set corresponding to the non-zero entries of $\widehat{M}^\top \widehat{M}$. Then, the following must be true:*

$$\mathbb{E}\left[\widehat{T}_{i,j}\middle| (i,j) \in \Omega\right] = T_{i,j}, \tag{2}$$

*for any $1 \le i, j \le d$. As a corollary of equation (2), we have that*

$$\mathbb{E}\left[\widehat{T}_{i,j}\right] = \Pr\left[(i,j) \in \Omega\right] \cdot T_{i,j}.$$

Unlike conventional results showing that the Hájek estimator is asymptotically consistent (Särndal et al., 2003), in the one-sided matrix completion setting, we find that the Hájek estimator is unbiased in the presence of finitely many samples. As a remark, the same unbiased estimation result can also be stated in the setting where every row has $k$ uniformly random entries from $M$, for arbitrary integers $k \ge 2$. The proof is similar to that of Lemma 3.1 and is omitted.

We now describe the main ideas for proving Lemma 3.1. First, we note that for any $k > 1$, conditioned on $(I^\top I)_{i,j} = k$, $\widehat{M}^\top \widehat{M}$ must be equal to the sum of $k$ pairs chosen from $M_{1,i}M_{1,j}$, $M_{2,i}M_{2,j}$, ..., $M_{n,i}M_{n,j}$. Furthermore, the choice of $k$ such pairs out of $n$ is uniformly at random among all possible $\binom{n}{k}$ combinations, because the choices are symmetric. Thus, the average of $k$ randomly chosen pairs must be equal to the average of all $n$ pairs in expectation. This is stated precisely in the following equation:

$$\mathbb{E}\left[\frac{1}{k}\left(\widehat{M}^\top \widehat{M}\right)_{i,j}\;\middle|\;(I^\top I)_{i,j} = k\right] = \frac{1}{n}\sum_{a=1}^{n} M_{a,i}M_{a,j} = T_{i,j}, \tag{3}$$

which leads to the conclusion of equation (2). See the full proof in Appendix A.

### 3.1 Variance Reduction

Having established that $\widehat{T}$ provides an unbiased estimator on the entries of $T$, we turn to studying the variance of $\widehat{T}$. We show that the Hájek estimator incurs lower variance relative to the Horvitz-Thompson estimator. We first derive an approximation of $Var(\widehat{T})$ to tackle the nonlinearity of the Hájek estimator.

**Theorem 3.2.** *Suppose $p = \frac{C}{d}$ for some fixed integer $C \ge 2$. Suppose $n \ge O(d \log d)$. Then, with probability at least $1 - O(d^{-1})$, (i) all of the diagonal entries of $\widehat{T}$ are in $\Omega$, and (ii) for any diagonal entries of $\widehat{T}$, the variance of $\widehat{T}_{i,i}$, for any $i = 1, 2, \ldots, d$, is approximated by*

$$Var\left(\widehat{T}_{i,i}\right) = \frac{1-p}{np}\left(\frac{1}{n}\sum_{k=1}^{n} M_{k,i}^4\right) - \frac{1-p}{np}T_{i,i}^2 + O\left(\frac{1}{np}\right). \tag{4}$$

*For any nonzero, off-diagonal entries in $\Omega$, the variance of $\widehat{T}_{i,j}$, for any $1 \le i \ne j \le d$, is approximated by*

$$Var\left(\widehat{T}_{i,j}\;\middle|\;(i,j) \in \Omega\right) = \frac{1}{n}\left(\sum_{k=1}^{n} M_{k,i}^2 M_{k,j}^2\right) - T_{i,j}^2 + O\left(\frac{(\log d)^2}{d}\right). \tag{5}$$

We now compare the variance of the Hájek estimator, $Var(\widehat{T})$, with the variance of the Horvitz-Thompson estimator, $Var(\overline{T})$, since both estimators are unbiased. We first derive the variance of $\overline{T}$ to make the comparison. For diagonal entries, we have that $\overline{T}_{i,i} = (np)^{-1}(\widehat{M}^\top \widehat{M})_{i,i}$, for $i = 1, \ldots, d$. For off-diagonal entries, we have $\overline{T}_{i,j} = (np^2)^{-1}(\widehat{M}^\top \widehat{M})_{i,j}$, for $i \ne j$. Then, we calculate the variance of $\overline{T}$ as:

$$Var\left(\overline{T}_{i,i}\right) = \frac{1-p}{n^2 p}\sum_{k=1}^{n} M_{k,i}^4, \ \forall\, i = 1, \ldots, d, \tag{6}$$

$$Var\left(\overline{T}_{i,j}\right) = \frac{1-p^2}{n^2 p^2}\sum_{k=1}^{n} M_{k,i}^2 M_{k,j}^2, \ \forall\, j = 1, \ldots, d \text{ and } j \ne i, \tag{7}$$

which can be verified based on the definition of $\overline{T}$ in Section 2. Now we can compare equations (6) and (7) with equations (4) and (5), respectively.

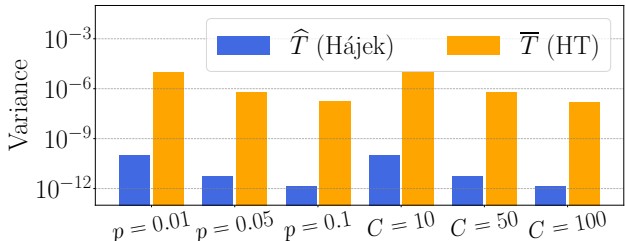

Figure 1: An illustration of the variance reduction using the Hájek estimator vs. the Horvitz-Thompson (HT) estimator, measured on synthetic data with $n = 10^4$ and $d = 10^3$, by repeating the data sampling procedure 100 times and calculating the variance across either uniform sampling with probability $p$, or sampling $C$ entries per row. In particular, $Var(\widehat{T})$ is generally $10^{-3}$ lower than $Var(\overline{T})$. For details regarding the simulation setup, see Section 5.

- Notice that the diagonal entries are nonzero with high probability over all the diagonal entries. By comparing equation (4) to equation (6), we see a reduction in the second term of equation (4), $-(1-p)T_{i,i}^2/(np)$, which is at the same order as the first term.

- The off-diagonal entries are nonzero with probability equal to $q \approx np^2 = \tilde{O}(d^{-1})$. By comparing equation (5) to (7), we see that equation (7) is larger than the first term of equation (5) by a factor of $np^2/(1-p^2)$. In addition, the second term of equation (5), which is always negative, further reduces to the variance of $\widehat{T}_{i,j}$ compared with $\overline{T}_{i,j}$.

*Algorithmic implications.* One consequence of variance reduction is that the empirical performance of the Hájek estimator tends to be much more stable than the Horvitz-Thompson estimator in the ultra-sparse setting. This is illustrated in Figure 1, measured on synthetic datasets with $n = 10^4$ and $d = 10^d$, for various values of $p$ (uniform sampling) and $C$ (i.e., sampling $C$ entries per row without repetition).

### 3.2 Sample Complexity

Next, we consider the entries outside $\Omega$. We minimize a reconstruction error objective between $\widehat{T}$ and $XX^\top$, for some $X \in \mathbb{R}^{d \times r}$, plus a regularization penalty of $R(X)$, as follows:

$$\ell(X) := \frac{1}{2} \left\| P_\Omega(XX^\top - \widehat{T}) \right\|_F^2 + \lambda R(X), \tag{8}$$

where $\lambda > 0$ is a regularization parameter. $R(X)$ is defined as $\sum_{i=1}^d (\|X_i\| - \alpha)_+^4$, where $X_i \in \mathbb{R}^r$ is the $i$-th row vector of $X$, for $i = 1, \ldots, d$, and $\alpha$ is a scalar that roughly corresponds to the row vector norms of $X$. This penalty regularizes the maximum $\ell_2$-row-norm of $X$, and is shown to provide theoretical properties in the optimization landscape of matrix completion (Ge et al., 2016; 2017). We summarize our procedure in Algorithm 1, which uses gradient descent to minimize $\ell(X)$. As a remark, in the loss objective $\ell(X)$, we recover the diagonal and off-diagonal entries of $\widehat{T}$ together via $XX^\top$, while reweighting the diagonal entries of $XX^\top$, based on the calculation in Lemma 3.1.[1]

We analyze this algorithm in a common means model, where each row of $M$ follows a mixture of $r$ vectors $u_1, u_2, \ldots, u_r \in \mathbb{R}^d$. Let $M_i$ denote the $i$-th row vector of $M$. For every $i = 1, 2, \ldots, n$, assume that $M_i$ is drawn uniformly at random from $u_1, u_2, \ldots, u_r$. Let $U = [u_1, u_2, \ldots, u_r]/\sqrt{r}$ denote a $d$ by $r$ matrix corresponding to the combined rank-$r$ factors. Let $\kappa = \sigma_{\max}(U)/\sigma_{\min}(U)$ denote the condition number of $U$. A critical condition to ensure guaranteed recovery in matrix completion is the following assumption regarding the row norms of $U$.

---

[1]We remark that there are other algorithms besides gradient descent for the imputation step, such as subspace recovery on the Grassmann manifold (Boumal & Absil, 2011). Zilber & Nadler (2022) design an iterative Gaussian-Newton algorithm for matrix recovery based on a linearization of the recovery objective. A natural extension of HÁJEK-GD involves using the Hájek estimator in the first step, and then applying these methods (instead of stochastic gradients) in the second step.

---

**Algorithm 1** Hájek estimation with gradient descent (HÁJEK-GD) for one-sided matrix completion

---

**Input:** A partially-observed data matrix $\widehat{M} \in \mathbb{R}^{n \times d}$
**Require:** Rank of the second-moment matrix $r$, number of iterations $t$, and learning rate $\eta$
**Output:** A $d$ by $d$ matrix

1: $I \in \mathbb{R}^{n \times d} \leftarrow$ The 0-1 indicator mask corresponding to the nonzero entries of $\widehat{M}$
2: $\Omega \subseteq [d] \times [d] \leftarrow$ The set of indices corresponding to the nonzero entries of $I^\top I$
3: $\widehat{T} \in \mathbb{R}^{d \times d} \leftarrow$ The element-wise division between $\hat{M}^\top \hat{M}$ and $I^\top I$ on $\Omega$
4: $X_0 \in \mathbb{R}^{d \times r} \leftarrow$ A random Gaussian matrix whose entries are sampled independently from $\mathcal{N}(0, d^{-1})$
5: **for** $i = 1, \ldots, t$ **do**
6: $\quad X_i \leftarrow X_{i-1} - \eta \nabla \ell(X_{i-1})$, where $\ell(\cdot)$ is defined in equation (8)
7: **end for**
8: **return** $\widehat{T}$ plus the entries of $X_t X_t^\top$ outside $\Omega$

---

**Assumption 3.3** (See also Definition 1.1 in Recht (2011) and Assumption 1 in Ge et al. (2016))**.** *Let $U e_i$ denote the $i$-th row vector of a $d$ by $r$ matrix $U$, where $e_i \in \mathbb{R}^d$ is the $i$-th basis vector, for any $i = 1, 2, \ldots, d$. The coherence of $U$ is given by*

$$\mu := \frac{d}{r} \max_{1 \leq i \leq d} \frac{\|U e_i\|^2}{\|U\|_F^2}. \tag{9}$$

Assuming that $\mu(U)$ is a fixed value that does not grow with $d$; That is, $\mu(U)$ does not increase with the dimensionality of the problem. We show the following performance guarantee of gradient descent for recovering $T$, measured in terms of the Frobenius norm distance between a local minimizer and $T$. In particular, recall that $\lambda$ refers to the strength of the regularization penalty, while $\alpha$ refers to the magnitude of the norm of the row vectors. See also the description of the incoherence penalty term below equation (8).

**Theorem 3.4.** *Suppose the rows of $M$ follow a mixture of $r$ common factors given by $U \in \mathbb{R}^{d \times r}$. Additionally, the coherence of $U$ is at most $\mu$ for some fixed $\mu \geq 1$ that does not grow with $d$. Suppose each entry of $M$ is observed with probability $p = \frac{C}{d}$ for some fixed integer $C \geq 2$, and let $q = 1 - (1 - p^2)^n$. Let $\alpha = 4\kappa^2 r \sqrt{\frac{\mu}{d}}$ and $\lambda = \frac{(r+1)dq}{16r^2\mu^3}$. When $n \geq \frac{cdr^5\kappa^6\mu^2 \log(d)}{C^2\epsilon^2}$ for some fixed constant $c$ and $\epsilon \in (0,1)$, with probability at least $1 - O(d^{-1})$ over the randomness of $\Omega$, when $d$ is large enough, any local minimizer $X$ of $\ell(\cdot)$ satisfies*

$$\left\| X X^\top - T \right\|_F^2 \leq \epsilon^2. \tag{10}$$

This result implies that any local minimum solution of the loss objective is also approximately a global minimum solution. We defer a proof sketch of this result to Section 4. We remark that prior results have established one-sided matrix completion guarantees when each row has two observed entries (Cao et al., 2023). By contrast, Theorem 3.4 applies to arbitrary values of $C \geq 2$ in each row.

Notice that Theorem 3.4 crucially relies on the assumption that $n$ is much larger than $d$. To provide several concrete examples, Table 2 in Section 5 presents the values of $n$ and $d$ for several real-world datasets, including MovieLens and Amazon reviews. In these examples, the average $n/d$ is roughly 10, which justifies the assumption that $n$ can be much larger $d$ in practice. Our experiments, presented later in Section 5.2, further validate the sample complexity result using synthetic data.

*Algorithmic implications.* A crucial step in the proof is to leverage the incoherence assumption, which reduces local optimality conditions from finite samples to the entire population. Later in Section 5.3, we also empirically demonstrate that the use of the incoherence regularization penalty helps improve performance on real-world datasets.

**Remark 3.5.** *The common means model is often used to study learning from heterogeneous datasets such as multitask learning and distributed learning. See, e.g., Kolar et al. (2011); Dobriban & Sheng (2020). In particular, Kolar et al. (2011) study union support recovery under the common means model, further assuming*

*that every mean vector is perturbed by another noise addition of $\epsilon_i$ (added to $M_i$). Despite the simplicity of this model, several learning problems can be reduced to the Normal means model (Brown & Low, 1996). Dobriban & Sheng (2020) study a distributed ridge regression problem, where each every individual performs a ridge regression on a local machine, and then, the ridge estimators are aggregated together on a central server. In particular, the samples of each individual are assumed to follow a linear model with the same $\beta$, plus independent random noise $\epsilon$ with mean zero and variance $\sigma^2$. One way to extend the common means model is by positing a noise addition, such as a noise vector $\epsilon_i$ added to $M_i$, whose entries are drawn independently from a Gaussian distribution $\mathcal{N}(0, \sigma^2/d)$. Our proof can be naturally applied to this extension. In particular, there are three places that need to be modified to account for the noise addition, including: 1) The proof of Lemma B.1, in particular equations (27), (28), and (29). 2) The proof of Corollary B.2, in particular, adding the dependence of $\sigma^2$ to the numerator in equations (31) and (32). 3) The proof of Corollary B.3, which inherits the dependence on $\sigma^2$ from Corollary B.2. These extensions will rely on concentration bounds on Gaussian random variables. Finally, we would also need to incorporate the noise variance in the diagonal entries of $\widehat{T}$. We can de-bias this by subtracting a suitably scaled diagonal matrix in the objective $\ell(X)$.*

*Another plausible extension is to instead assume that each $M_i$ is a linear combination of the $r$ low-rank factors. This setting appears to be significantly more challenging and would likely go beyond the techniques developed in this paper, as one would first need to disentangle the linear dependence and design a new estimator capable of learning the low-rank subspace spanned by the latent factors. One promising direction for tackling this problem is to use the spectral estimator developed in the work of Chen et al. (2021). This is left as an open question for future work.*

**Discussions.** In the proof of Theorem 3.4, we tackle the error terms arising from the diagonal and off-diagonal entries separately. This is because the diagonal entries are fully observed with high probability. By contrast, the off-diagonal entries are only sparsely observed. At a high level, as $n$ increases, the concentration errors in the diagonal entries will decrease (at a rate of $n^{-1/2}$). Meanwhile, $q$ will also increase, leading to more observed samples in the off-diagonal entries. Thus, we can set $n$ carefully to ensure that the error between $\widehat{T}$ and $T$, when restricted to $\Omega$, is sufficiently small. The details can be found in Appendix B.1-B.2.

We now discuss our work from the perspective of prior work by Fan et al. (2013). Their work examines a low-rank plus noise common factor model. Their estimators involve a low-rank estimator, plus a diagonal matrix. In particular, this diagonal matrix is designed to offset the bias contributed by the noise term. As an extension, if we incorporate the noise term into the latent factor model of $M_i$, we would similarly need to de-bias the diagonal entries to account for the norms of the noise. The proof can be extended to handle this setting by estimating the magnitude of the noise variance, and then adding this bias to the objective $\ell(X)$.

## 4   Proof Techniques and Extensions

Next, we present a sketch of our proof. There are two major challenges in analyzing the Hájek estimator in the ultra-sparse sampling regime: First, the estimator involves dividing one random variable by another random variable, resulting in a nonlinear estimator. Further, the missing patterns in $\Omega$ are non-random, depending on the diagonal and off-diagonal entries, respectively. Our approach to tackle this non-linearity is via a first-order approximation technique for analyzing the Hájek estimator on the diagonal entries. Further, we carefully analyze the bias in the off-diagonal entries.

Second, the sampling patterns of $\Omega$ are not fully independent. To this end, we establish a concentration inequality that handles the concentration error row by row in $\Omega$. This is based on the observation that, conditioned on one row, expect the diagonal entries, the randomness of each entry in that row would be independent across different entries. This row-by-row concentration argument also allows us to analyze the spectral norm of the bias matrix. As a remark, this type of argument has been used in matrix completion with non-random missing data (Athey et al., 2021). However, prior work focuses on analyzing nuclear norm regularization methods, while our result now applies to gradient-based optimization. Next, we outline the main results in addressing the above two challenges.

### 4.1 Main Proof Ideas

*Bias and variance of the Hájek estimator.* Recall that $\widehat{T}$ is unbiased by Lemma 3.1. Thus, the variance of $\widehat{T}$ is the same as the expected squared error between $\widehat{T}$ and $T$. A key result for deriving the variance of the Hájek estimator is as follows.

**Lemma 4.1.** *In the setting of Theorem 3.4, suppose $p = C/n$ for some fixed integer $C \geq 2$ and $n \geq d(\log d)/\epsilon^2$. Let $S_i = \{j : (i,j) \in \Omega, j \neq i\}$ denote the set of indices corresponding to the nonzero entries in the $i$-th row of $\widehat{T}$. With probability at least $1 - O(d^{-1})$, the following must be true, for every $i = 1, 2, \ldots, d$,*

$$Var\left(\widehat{T}_{i,i}\right) \leq \frac{\mu^2 r}{d^2 np} \left(1 + \epsilon\sqrt{\frac{6C^{-1}}{d}}\right) \left(1 + r\sqrt{\frac{3\log(d/2)}{2n}}\right), \tag{11}$$

$$\sum_{j \in S_i} Var\left(\widehat{T}_{i,j}\right) \leq \frac{\mu q}{d} \left(1 + 3r\sqrt{\frac{3\log(d/2)}{2d}}\right) + \tilde{O}\left(d^{-3}\right). \tag{12}$$

*Proof sketch.* The proof regarding the diagonal entries in equation (11) is based on a first-order approximation of $Var(\widehat{T}_{i,i})$. For every $i = 1, 2, \ldots, d$, let

$$A_{i,i} := \sum_{k=1}^{n} M_{k,i}^2 I_{k,i} \quad \text{and} \quad B_{i,i} := \sum_{k=1}^{n} I_{k,i}. \tag{13}$$

We perform Taylor's expansion of $\widehat{T}_{i,i}$ around $(\mathbb{E}\left[A_{i,i}\right], \mathbb{E}\left[B_{i,i}\right])$ as follows (See also Chapter 5.5, Särndal et al. (2003)):

$$\widehat{T}_{i,i} = \frac{\mathbb{E}\left[A_{i,i}\right]}{\mathbb{E}\left[B_{i,i}\right]} + \frac{1}{\mathbb{E}\left[B_{i,i}\right]}(A_{i,i} - \mathbb{E}\left[A_{i,i}\right]) - \frac{\mathbb{E}\left[A_{i,i}\right]}{(\mathbb{E}\left[B_{i,i}\right])^2}(B_{i,i} - \mathbb{E}\left[B_{i,i}\right]) + \epsilon_{i,i}. \tag{14}$$

Above, the first term to the right of equation (14) stems from the zeroth-order expansion. The second and third terms arise from taking the partial derivatives over $A_{i,i}, B_{i,i}$, respectively. $\epsilon_{i,i}$ is an error term that is at the order of the variance of $A_{i,i}, B_{i,i}$, which can be calculated in close form based on equation (13).

Given that $\widehat{T}$ is unbiased, $Var(\widehat{T}_{i,i})$ is equal to the expectation after squaring both sides of equation (14). As a result, we derive the variance approximation of $\widehat{T}_{i,i}$ by carefully analyzing each term, which will lead to equation (4).

- Figure 2 provides an illustration of equation (14) (except the $\epsilon_{i,i}$ error term) and the corresponding variance estimate from equation (4). We find that both approximations hold with negligible errors for various sampling probabilities $p$ (uniform sampling with probability $p$) and $C$ (fixed number of entries per row).

- This simulation uses $n = 10^4$ and $d = 10^3$ and follows the setup stated in the synthetic data generation process, which can be found in Section 5.

As for the off-diagonal entries of $\widehat{T}$, note that when $(i,j) \in \Omega$, with high probability the $(i,j)$-th entry of $I^\top I$ must be one. Thus, $Var(\widehat{T}_{i,j})$ is equal to the variance of $n$ values $M_{1,i}M_{1,j}, \ldots, M_{n,i}M_{n,j}$ (plus some small errors). This leads to the variance approximation of equation (5). See Appendix B.1 for the full proof.

*A concentration inequality for operator $P_\Omega$.* In order to analyze the optimization landscape of the loss objective $\ell(\cdot)$, we will first derive the first-order and second-order optimality conditions. However, these are stated on the observed set $\Omega$, and we need to turn them into the full set on $T$. To this end, we develop the following concentration inequality, which helps reduce finite-sample local optimality conditions to the full matrix.

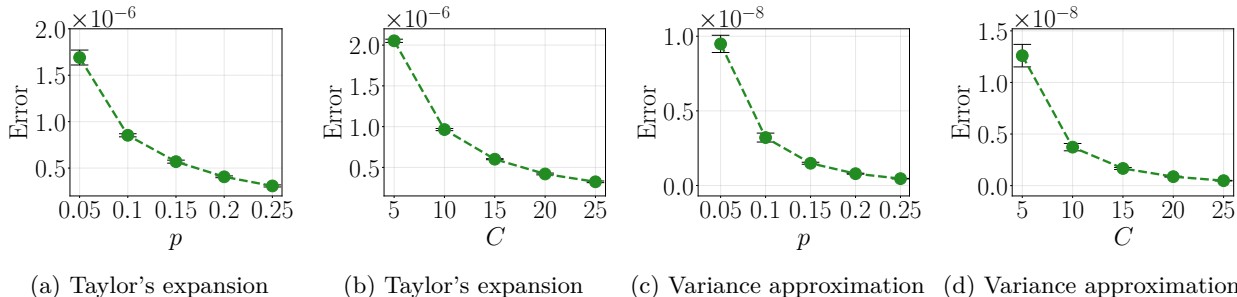

(a) Taylor's expansion  (b) Taylor's expansion  (c) Variance approximation  (d) Variance approximation

Figure 2: We illustrate the results from applying a first-order approximation to the variance of the diagonal entries of HÁJEK-GD, run on synthetic data with $n = 10^4$ and $d = 10^3$. Figures 2a and 2c: Sample each entry with probability $p$. Figures 2b and 2d: Sample $C$ entries per row without repetition. In particular, the approximation errors incurred by both first-order Taylor's expansion and the variance approximation are generally less than $10^{-6}$.

**Lemma 4.2.** *Let $W \in \mathbb{R}^{d \times d}$ be a d by d symmetric matrix such that $\|W\|_\infty \leq \frac{\nu}{d}$, for some fixed positive value of $\nu$. Then, with probability at least $1 - O(d^{-1})$, the following statement must be true:*

$$\|P_\Omega(W) - qW\|_2 \leq q\nu \sqrt{\frac{16 \log(2d)}{dq}}. \tag{15}$$

The proof of this concentration result can be found in Appendix B.2.

The last ingredient involves analyzing the population loss of one-sided matrix completion, building on the machinery of Ge et al. (2016). These are discussed in detail in Appendix B.3 and B.4, which involve analyzing the population landscape and then reducing empirical case to the population case, respectively.

- In particular, we extend their analysis to account for the bias of $\widehat{T}$ and the non-random missingness of $\Omega$ due to the diagonal and off-diagonal entries.

- The treatment of the bias of $\widehat{T}$ relative to $T$ is especially tedious and requires careful calculation. We build on Lemma 4.1 and 4.2 to give the bias of each diagonal entry, and the bias of each row — See Corollary B.2.

- The final proof of Theorem 3.4 can be found in Appendix B.5. We also discuss our work from the perspective of the work of Ge et al. (2016) in Remark B.12.

**Remark 4.3.** *The above sample complexity result assumes the existence of a low-rank factor model, and the gradient descent algorithm requires knowing the exact rank of the factor model. In practice, one can adjust the rank parameter in the gradient descent algorithm via cross-validation. See a detailed ablation analysis in Appendix C.3. It is an interesting question to examine adaptive optimization methods that can automatically learn or estimate the true rank without knowing it. One natural extension of our technique is by running SVD on $\widehat{T}$ and from the decomposition infer the rank of $T$. For example, this can be achieved by analyzing the spectrum induced by the noise added to $\widehat{T}$. See Corollary B.2 in Appendix B.1 for the analysis of the bias (i.e., $\widehat{T} - T$) on $\Omega$.*

## 4.2 Extensions

We now describe two extensions of our approach. In the first extension, we use HÁJEK-GD to recover the unobserved entries of $M$. To accomplish this, we solve a least-squares regression problem that projects the observed entries of $M$ onto the span of the recovered second-moment matrix. An illustration of this overall procedure is provided in Figure 3, and the complete algorithm is detailed in Algorithm 2. In particular, when $C \geq O(r \log d)$, one can show that this least-squares procedure can accurately recover every row of $M$.

---

**Algorithm 2** Missing data imputation using one-sided matrix completion

---

**Input:** A partially observed $\widehat{M} \in \mathbb{R}^{n \times d}$
**Require:** Rank $r$, number of iterations $t$, learning rate $\eta$
**Output:** An $k$ by $d$ matrix corresponding to the imputed rows of $S$
1: $Z \leftarrow$ HÁJEK-GD$(\widehat{M}; r, t, \eta)$           // May add Gaussian noise to the non-zero entries of $\widehat{M}$
2: $U_r D_r U_r^\top \leftarrow$ rank-$r$ SVD of $Z$
3: $\Omega \leftarrow$ set of indices corresponding to the nonzero entries of $\widehat{M}^\top \widehat{M}$
4: $Q \leftarrow \arg\min_{Q \in \mathbb{R}^{n \times r}} \frac{1}{2} \sum_{i \in S, (i,j) \in \Omega} \left( (QU_r^\top)_{i,j} - \widehat{M}_{i,j} \right)^2$
5: **return** $QX^\top$

---

As discussed in the introduction, recovery guarantees are generally impossible when the number of samples is extremely limited. However, one practical way to view this algorithm is as a user-level recovery procedure: when a user performs the local least-squares regression, it is likely that they will have access to additional local data beyond the ultra-sparse global observations. In such cases, the least-squares step can accurately reconstruct the entire row based on the recovered row space.

In the second extension, we further generalize our recovery guarantees to account for differential privacy (DP) using the Gaussian mechanism. We begin by recalling the notion of differential privacy in the context of matrix completion. Given an input $\widehat{M}$, an algorithm $\mathcal{A}$ is said to satisfy $(\varepsilon, \delta)$-joint differential privacy if, for any $i \in \{1, 2, \ldots, n\}$, replacing the $i$-th row of $M$ (and the corresponding row of $\widehat{M}$) by another vector $x \in \mathcal{X} \subseteq \mathbb{R}^d$ yields matrices $\widehat{M}$ and $\widehat{M}'$ that differ in only one row. Then, for any set of measurable events $S$,

$$\Pr\left[\mathcal{A}(\widehat{M}) \in S\right] \leq \exp(\varepsilon)\Pr\left[\mathcal{A}(\widehat{M}') \in S\right] + \delta, \tag{16}$$

This definition follows prior work on user-level DP for matrix completion (Liu et al., 2015; Wang et al., 2023). To satisfy $(\varepsilon, \delta)$-DP, we perturb the nonzero entries of $\hat{M}$ with Gaussian noise drawn from $\mathcal{N}(0, \sigma^2)$. To determine the appropriate noise level $\sigma$, we measure the $\ell_2$-sensitivity of the algorithm $\mathcal{A}: \mathbb{R}^{n \times d} \to \mathbb{R}^{d \times d}$ as:

$$\Delta_2(\mathcal{A}) = \max_{M \sim M'} \|\mathcal{A}(M) - \mathcal{A}(M')\|_F, \tag{17}$$

where $M \sim M'$ denotes that the two matrices differ in at most one row.

By standard results in the DP literature, when

$$\sigma = \frac{2\sqrt{\ln(1.25\delta^{-1})}\Delta_2(\mathcal{A})}{\varepsilon},$$

the algorithm HÁJEK-GD combined with the Gaussian mechanism satisfies the $(\varepsilon, \delta)$-joint differential privacy with high probability. See Theorem A.1 in Dwork & Roth (2014) for the full statement. In Appendix C.3, we empirically estimate the sensitivity of HÁJEK-GD and find that $\Delta_2(\mathcal{A})$ remains small—typically between 11 and 19 across several datasets.

Together, these two extensions show that our approach can (i) leverage the recovered row space to impute missing entries of $M$, and (ii) ensure privacy-preserving estimation under standard Gaussian noise perturbation.

## 5 Experiments

In this section, we evaluate our proposed approach through extensive experiments on both synthetic data and real-world datasets.

**Synthetic data generation.** We generate synthetic data by sampling each entry of $M \in \mathbb{R}^{n \times d}$ independently from a Gaussian distribution $\mathcal{N}(1/\sqrt{d}, 1/d)$, where $d$ denotes the number of columns. Unless otherwise stated, we fix $n = 10^4$ and $d = 10^3$. To construct a low-rank ground-truth matrix, we perform an SVD of $M$ and

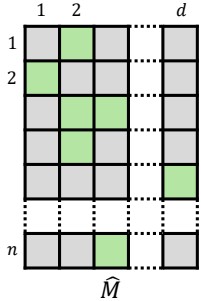 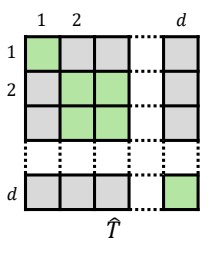 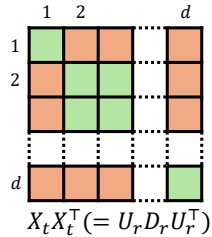 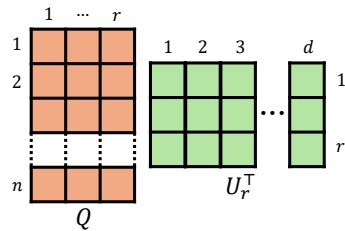

**Input**: A partially-observed data matrix   **Step 1:** Using Hájek estimator   **Step 2:** Impute missing entries of $T$ using gradient descent   **Extension:** Impute missing entries of $M$ using least squares projection

Figure 3: *Illustration of the proposed one-sided matrix completion pipeline.* **Input:** A partially observed matrix $\hat{M} \in \mathbb{R}^{n \times d}$. Each row represents a user's data (e.g., movie ratings), and each column represents an item. Thus, $M$ is a tall-and-skinny matrix, with $n$ being larger than $d$. **Step 1:** Apply the Hájek estimator to correct for non-random missingness in the empirical second-moment matrix $\hat{M}^\top \hat{M}$ over the observed index set $\Omega$. **Step 2:** Impute the remaining unobserved entries of $T$ by running gradient descent on a low-rank reconstruction loss between the estimated and true second-moment matrices. The final $T$ combines the observed entries from $\widehat{T}$ with the missing entries from $X_t X_t^\top$, where $X_t$ denotes the $t$-th gradient descent iterate. **Extension:** Perform full matrix imputation by projecting onto the recovered low-rank subspace $U_r$ (obtained via rank-$r$ SVD of the estimated $T$) and solving a least-squares regression problem to estimate the missing entries of $M$.

retain the top-10 singular values and corresponding singular vectors, truncating the lower tail. The observed matrix $\hat{M}$ is then obtained by sampling each entry independently with probability $p = C/d$. We also consider an alternative sampling scheme that selects exactly $C$ entries per row without replacement, and find that the results are qualitatively similar under both schemes.

**Real-world datasets.** We further evaluate our algorithm on several large, sparse matrix datasets: (i) three MovieLens datasets with 20 million and 32 million nonzero entries; (ii) an Amazon Reviews dataset (Hou et al., 2024); and (iii) a Genomics dataset from the 1,000 Genomes Project (Cao et al., 2023). These datasets are standard testbeds for sparse matrix completion and cover a wide range of sparsity leves and scales representative of real-world recommendation and biological data. Their detailed statistics are summarized in Table 2. We formulate the prediction task on each dataset as a regression problem.

**Baselines.** We compare our method against three strong baselines widely used in the matrix-completion literature:

1. Nuclear-norm regularization (Cao et al., 2023), which coincides with the same weighted estimator as HÁJEK-GD when each row contains exactly two entries. One difference between our implementation and theirs is that we also add the incoherence regularization term to the loss function.

2. Alternating Gradient Descent (alternating-GD).

3. Soft-Impute with Alternating Least Squares (softImpute-ALS) (Hastie et al., 2015).

To adapt the latter two methods for one-sided matrix completion, we first apply each baseline to recover the full matrix $M$ and then compute the corresponding second-moment matrix $T$ as $M^\top M/n$. Additional dataset descriptions and implementation details are provided in Appendix C.

**Evaluation metrics.** For one-sided matrix completion, we measure compare the recovery error as the Frobenius norm between the estimated and true $T$ matrices. For full matrix imputation, we compare Algorithm 2 to alternating-GD and softImpute-ALS, reporting the root mean squared error (RMSE) between the imputed and ground-truth values. All experiments are implemented in PyTorch and executed on an Ubuntu server with 16 Intel Xeon CPUs and one Nvidia Quadro 6000 GPU.

Table 2: The detailed statistics of five real-world matrix datasets used in our experiments.

| Dataset | Genomes | Amazon Reviews | MovieLens-20M | MovieLens-25M | MovieLens-32M |
|---|---|---|---|---|---|
| # Rows | $100,000$ | $100,000$ | $138,000$ | $162,000$ | $200,948$ |
| # Columns | $2,504$ | $50,000$ | $27,000$ | $62,000$ | $87,585$ |
| # Nonzero | $2.5 \times 10^8$ | $1.3 \times 10^6$ | $2 \times 10^7$ | $2.5 \times 10^7$ | $3.2 \times 10^7$ |

**Summary of numerical results.** For one-sided matrix completion in the sparsest regime—where each row has only two observed entries—we find that HÁJEK-GD reduces recovery error by 70% compared to nuclear-norm regularization on synthetic data. Across all three real-world datasets (each with sparsity below 1%), HÁJEK-GD reduces the recovery error of $T$ by at least 42% relative to alternating-GD and by 59% relative to softImpute-ALS.

When using HÁJEK-GD to impute the missing entries of $M$ as part of Algorithm 2, the overall approach achieves an 85% reduction in RMSE compared to alternating-GD and softImpute-ALS on synthetic data where each row has only two entries. On real-world datasets, HÁJEK-GD further reduces RMSE by at least 21% relative to alternating-GD and by 38% relative to softImpute-ALS.

Finally, we conduct ablation studies to validate our algorithmic design choices. In particular, we highlight the benefits of incorporating the incoherence regularization $R(X)$ in the loss and demonstrate that HÁJEK-GD remains robust under random noise perturbations in the input. Together, these findings indicate that **our approach is particularly effective and stable in ultra-sparse matrix regimes**.

## 5.1 Measuring Bias on Observed Entries

We begin by comparing the bias between $\widehat{T}$ and $\overline{T}$, measured by the sum of squared errors between $\widehat{T}$ (and $\overline{T}$) with $T$ on $\Omega$. Our finding is illustrated in Figure 4. First, we find that the bias of $\widehat{T}$ compared to $T$, is at the order of $10^{-4}$ to $10^{-2}$ for various $p$. In Figure 4a, we find that on synthetic datasets, the bias of $\widehat{T}$ is $9 \times 10^{-4}$ and the bias of $\overline{T}$ is $0.3$ averaged over various sampling probabilities, resulting in a reduction of **99**%. In Figure 4b, we observe that for three MovieLens datasets, the bias of $\widehat{T}$ is $6 \times 10^{-4}$ and the bias of $\overline{T}$ is $5 \times 10^{-3}$, resulting in a reduction of **88**%.

Additionally, we test a biased sampling procedure, where users who watch more movies tend to rate more movies as well (also known as "snowball effects" (Chen et al., 2020)). For each $i = 1, \ldots, n$, we sample its entries with probability $p_i$ equal to $C/d$ times the number of non-zeros in row $i$. We find that under this biased sampling procedure, the bias of $\widehat{T}$ is $3 \times 10^{-4}$ and the bias of $\overline{T}$ is $5 \times 10^{-3}$, resulting in a reduction of 93%. As explained earlier, these results stem from the variance reduction effect of $\widehat{T}$.

## 5.2 Recovery Results for Synthetic Matrix Data

We now examine one-sided recovery error by keeping the number of samples $m$ fixed while varying $p$ and $d$ separately. Since the sample complexity on $n$ grows linearly with $d$ times $\log(d)$, the error should remain roughly constant as we fix $m/d$ while slowly increasing $d$. We report simulation results under three regimes. For all three setting, we fix the number of rows $n = 10^4$.

First, we set the rank $r = 10$, noise injection variance $\sigma^2 = 1$, and vary $d$ from $10^3$ to $5 \times 10^3$ and $p$ from $2/d$ to $10/d$. As shown in Figure 5a, the estimation error remains flat for different values of $p$ and $m$, which suggests that sample complexity does not grow with $d$, after adjusting for $m/d$.

Second, we set the sampling probability $p = 2/d$, $\sigma^2 = 1$, and vary $d$ from $10^3$ to $5 \times 10^3$ and $r$ from 5 to 25. As shown in Figure 5b, the estimation error remains flat for a given rank $r$, as $d$ grows.

Third, we set the dimension $d = 10^3$, $p = 2/d$, $r = 10$, while varying $\sigma^2$ from 1 to 50. We find that as $\sigma^2$ increases, the estimation error grows linearly, as shown in Figure 5c. This is consistent with our bounds in Theorem 3.4, where $n$ grows proportionally with $\sigma^2$.

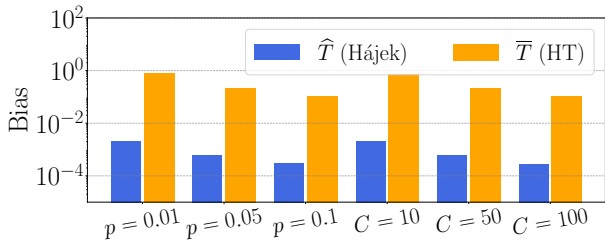

(a) Comparing the bias of $\widehat{T}$ with the bias of $\overline{T}$

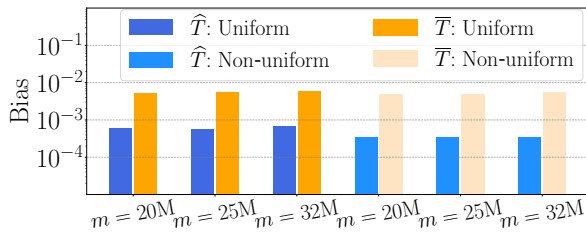

(b) Comparing the bias of $\widehat{T}$ with the bias of $\overline{T}$

Figure 4: Comparing the bias of the Hájek estimator vs. the Horvitz-Thompson estimator, measured as the mean squared error between $\widehat{T}$ (or $\overline{T}$) and $T$ on the index set $\Omega$. Figure 4a: Reporting the bias on synthetic matrix data with $n = 10^4$ and $d = 10^3$. Figure 4b: Reporting the bias on MovieLens datasets. We consider both uniform sampling with probability $p$ as well as sampling $C$ entries from each row. In Figure 4b, we further consider a non-uniform sampling setting, where the sampling probability at each row is proportional to the number of nonzero entries in that row.

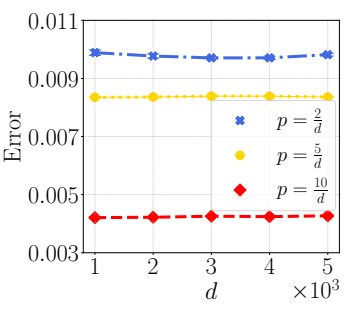

(a) Varying $d$ for different $p$

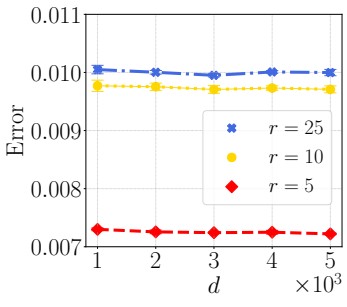

(b) Varying $d$ for different $r$

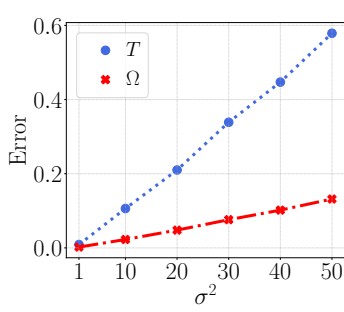

(c) Varying noise variance $\sigma$

Figure 5: Illustration of the recovery error of Algorithm 1. Figure 5a: We vary the sampling probability and find that the error roughly stays constant after we also fix the number of samples. Figure 5b: We see similar results as we fix the number of samples but vary the rank of the underlying matrix. Figure 5c: We gradually increase the noise level and find that the estimation errors also increase. The reported results are aggregated based on five independent runs.

Similar results on synthetic data are further illustrated in Figure 7 (See Appendix C.2). We see a consistent trend in which our approach reduces the recovery error across different settings of $C$ (expected number of entries at each row) and $n$, compared to all three baseline methods, including alternating-GD, softImpute-ALS, and nuclear norm regularization. In the setting with the fewest observed entries ($C = 2$ and $n = 10^4$), Hájek-GD achieves the lowest error of 0.10, representing a 70% reduction compared to the closest baseline, nuclear norm regularization, which yields an error of 0.48. In the setting with $C = 10$ observed entries per row, Hájek-GD achieves an error of 0.06, outperforming alternative-GD and nuclear norm regularization, whose error rates are 0.09 and 0.17, respectively.

## 5.3 Recovery Results for Real-World Matrix Data

Next, we report the results on three real-world datasets in Table 3. We find that our approach consistently achieves the lowest error for recovering $T$ and $M$, compared to two baseline methods. For one-sided matrix completion, Hájek-GD improves upon alternating-GD by 42% and softImpute-ALS by 59% on average. For recovering the entire matrix, our approach reduces recovery error by 21% relative to alternating-GD and by 38% relative to softImpute-ALS, averaged over the three datasets. In particular, on the Amazon reviews dataset, Hájek-GD reduces estimation of $T$ by **59**% and recovery of $M$ by **21**% relative to the best performing baseline.

Table 3: Results from applying our approach on three real-world datasets with up to 32 million entries, as compared with two baseline methods, namely alternating-GD and softImpute-ALS. We report both the recovery errors and the running time (measured in the number of seconds when evaluated on an Ubuntu server). We run each experiment with five random seeds to calculate the mean and the standard deviation.

| Dataset | MovieLens-32M | Amazon Reviews | Genomes | MovieLens-32M | Amazon Reviews | Genomes |
|---|---|---|---|---|---|---|
| $T$ | One-sided recovery error | | | Running time | | |
| AlternatingGD | $9.1_{\pm 0.1} \times 10^{-3}$ | $3.3_{\pm 1.3} \times 10^{-3}$ | $7.0_{\pm 1.6} \times 10^{-5}$ | $7.2_{\pm 0.1} \times 10^{2}$ | $4.8_{\pm 0.1} \times 10^{2}$ | $23.2_{\pm 5.0}$ |
| SoftImputeALS | $9.1_{\pm 0.1} \times 10^{-3}$ | $3.2_{\pm 0.3} \times 10^{-3}$ | $1.9_{\pm 0.5} \times 10^{-4}$ | $6.3_{\pm 4} \times 10^{4}$ | $5.1_{\pm 0.3} \times 10^{3}$ | $959.8_{\pm 40}$ |
| Algorithm 1 | $\mathbf{4.7_{\pm 0.1} \times 10^{-3}}$ | $\mathbf{1.3_{\pm 0.2} \times 10^{-3}}$ | $\mathbf{5.8_{\pm 0.1} \times 10^{-5}}$ | $\mathbf{1.5_{\pm 0.1} \times 10^{2}}$ | $\mathbf{2.8_{\pm 0.2} \times 10^{2}}$ | $\mathbf{1.3_{\pm 0.1}}$ |
| $M$ | Root mean squared recovery error | | | Running time | | |
| AlternatingGD | $1.7_{\pm 0.1}$ | $2.6_{\pm 0.1}$ | $0.2_{\pm 0.1}$ | $5.7_{\pm 0.1} \times 10^{2}$ | $2.2_{\pm 0.1} \times 10^{2}$ | $23.2_{\pm 5.0}$ |
| SoftImputeALS | $1.4_{\pm 0.1}$ | $3.3_{\pm 0.1}$ | $0.4_{\pm 0.1}$ | $6.3_{\pm 4} \times 10^{4}$ | $4.8_{\pm 0.3} \times 10^{3}$ | $959.9_{\pm 40}$ |
| Algorithm 2 | $\mathbf{1.1_{\pm 0.1}}$ | $\mathbf{1.9_{\pm 0.1}}$ | $\mathbf{0.2_{\pm 0.1}}$ | $\mathbf{5.6_{\pm 0.1} \times 10^{2}}$ | $\mathbf{2.8_{\pm 0.2} \times 10^{2}}$ | $\mathbf{14.7_{\pm 0.2}}$ |

An important design in HÁJEK-GD is the use of an incoherence regularization penalty of $R(\cdot)$ (see Section 3.2). We now describe the results from varying regularization parameter $\lambda$ and the threshold $\alpha$ (used in the truncation of $R(\cdot)$), as part of $R(\cdot)$, on a synthetic dataset with $p = 10/d$ and also the Amazon Review dataset. We vary $\lambda$ between $10^{-4}, 10^{-3}, 10^{-2}$, and we vary $\alpha$ between $10^{-5}, 10^{-4}, 10^{-3}, 10^{-2}, 10^{-1}$. We find that on synthetic datasets, the lowest recovery error is achieved when $\lambda = 10^{-4}$ and $\alpha = 10^{-3}$, leading to an error of $2.8 \times 10^{-3}$, compared to the error rate of $3.6 \times 10^{-3}$ without using the regularization. In other words, we can reduce the recovery error by 22%.

On the Amazon Reviews dataset, the lowest recovery error is achieved when $\lambda = 10^{-2}$ and $\alpha = 10^{-1}$, at $1.3 \times 10^{-2}$, compared to $2.7 \times 10^{-2}$ without using the incoherence regularizer. As a result, incorporating the incoherence regularization penalty of $R(\cdot)$ reduces the recovery error by 52%.

**Runtime scaling as the dimensions grow.** Below, we report experiment results to illustrate the runtime scaling of HÁJEK-GD as a function of both $d$ (number of columns) and $|\Omega|$ (number of nonzero entries in $\widehat{T}$). In particular, we consider both synthetic data and real-world data. In Figure 6a, we illustrate the running time (measured in seconds) as a function of the dimension $d$. We also vary $C$ to ensure the robustness of our findings. In Figure 6b, we illustrate the running time from HÁJEK-GD on the MovieLens datasets with various sampling probabilities $p$. We find that the runtime of HÁJEK-GD scales near linearly with the number of observed entries $|\Omega|$. It is an interesting question to precisely analyze the runtime scaling of gradient descent in low-rank matrix completion. See, e.g., several recent works for references (Li et al., 2020b; Ju et al., 2022; 2023; Ma & Fattahi, 2024; Zhang et al., 2024).

## 6 Related Work

The problem of recovering a low-rank matrix from a few potentially noisy entries has a rich history of studies in the literature, spanning a wide variety of areas, including collaborative filtering, machine learning, and compressed sensing (Candes & Plan, 2010). One of the earliest results from the matrix completion literature involves a maximum-margin matrix factorization approach (Srebro et al., 2004), which minimizes the reconstruction error plus a trace norm penalty on the low-rank factors. Under an incoherent condition on the low-rank factors plus a joint incoherence condition, Candes & Recht (2012) show that exact recovery of the unknown matrix is possible by minimizing the nuclear norm of the reconstructed matrix, subject to equality constraints on the observed entries. The joint incoherence condition on the factors is further shown to be unnecessary (Chen, 2015). Ding & Chen (2020) sharpen the known bounds through a leave-one-out analysis, which leads to a convergence guarantee for projected gradient descent on a rank-constrained formulation. Complementary to these results, Zhang et al. (2019b) show error bounds on the nuclear norm regularized objective, relative to the error of the best rank-$r$ approximation of $M$. Further references about different algorithmic approaches to matrix completion can be found in the survey by Sun (2015).

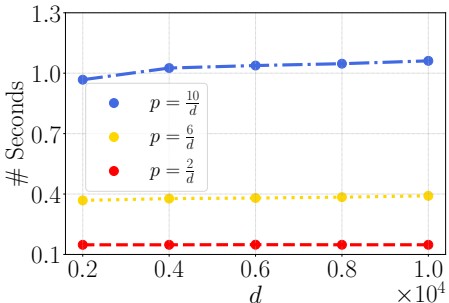
(a) Runtime on synthetic data by varying $d$

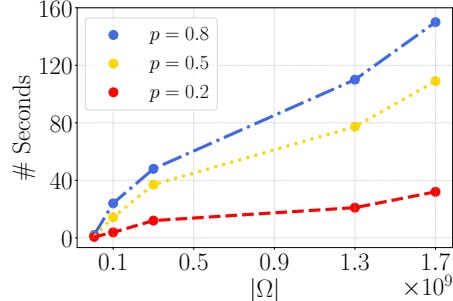
(b) Runtime on real-world data by varying $|\Omega|$

Figure 6: Illustration of the runtime scaling of HÁJEK-GD as $d$ and $m$ increase. Figure 6a: We vary the dimension $d$ and find that the runtime of HÁJEK-GD increases roughly linearly in $d$ as $d$ increases. To ensure the robustness of our findings, we report the results for three values of $C$ (recall that the sampling probability $p = C/d$). Figure 6b: We vary the number of entries $m$ between three MovieLens datasets and find that the runtime of HÁJEK-GD increases as $m$ increases. The reported results are based on five independent runs.

Recent literature has studied the optimization geometry of the loss landscape of matrix completion using rank-constrained first-order optimization methods (Sun & Luo, 2016; Ge et al., 2016). Provided with a regularization on the incoherence (balance between the rows of low-rank factors), it is possible to prove that all local minimum solutions of the matrix completion reconstruction error objective are also global minimum (Ge et al., 2017). By carefully analyzing the local geometry of the matrix factorization objective, it is possible to achieve exact recovery under the incoherence condition (Sun & Luo, 2016). Provided with this characterization, one can obtain convergence rates through the literature on finding local minima in nonconvex optimization problems (Nesterov & Polyak, 2006; Wang et al., 2019). In particular, it is known that many optimization algorithms, including cubic regularization, trust-region methods, and stochastic gradient descent, can efficiently find a local minimizer. See a recent textbook on first-order and stochastic optimization methods for further references (Lan, 2020).

There is a closely related problem of low-rank matrix recovery, whose aim is to reconstruct a low-rank matrix based on a linear system of measurement equations of the unknown matrix (Recht et al., 2010). The optimization landscape of low-rank matrix recovery from random linear measurements in the presence of arbitrary outliers is studied by Li et al. (2020a). Besides, there are studies on the dynamics of gradient descent assuming the factorization is over-parameterized (Li et al., 2018; Ma & Fattahi, 2024). Additionally, matrix factorization has connections to two-layer neural networks and random matrices, which have inspired studies on the dynamics of deep linear networks (Li et al., 2018). Another related problem is nonnegative matrix factorization, which has also been studied for "tall-and-skinny" matrices (Benson et al., 2014). There is also a line of work on the recovery of a low-rank plus sparse matrix (Hsu et al., 2011). Another plausible direction in the ultra-sparse sampling setting is to instead recover a low-rank approximation of the underlying matrix from just $O(n)$ samples. See Gamarnik et al. (2017) for further references of this setting.

The one-sided matrix completion problem has been recently formulated and studied in the special case where each row contains only two randomly observed entries (Cao et al., 2023). This work complements their paper in four aspects. First, their sample complexity bound focuses on the setting where every row has two observed entries, while our result applies to more general settings that allow for any constant $C$. Second, their result builds on proof techniques for matrix completion with nuclear norm regularization penalty. By contrast, our result applies to gradient-based optimization algorithms, which are faster and easier to implement in practice. Third, our result builds on the incoherence assumption from the MC literature. Finally, we explore the use of one-sided matrix completion for full matrix completion and find that this can also lead to improved results for imputing missing entries of the full matrix.

First-order stochastic gradient methods also have applications in large-scale matrix completion (Mackey et al., 2011). In particular, one can apply stochastic gradient updates in an asynchronous protocol, making it suitable for distributed platforms (Recht & Ré, 2013). Besides the nuclear norm minimization approach, alternating

minimization such as alternating least squares and low-rank matrix factorization are also widely used in practice (Hastie et al., 2015). More recently, Wang et al. (2023) consider a low-rank matrix factorization approach to private matrix completion. Assuming the $M$ matrix is indeed of low rank, their work shows a sample complexity bound that scales linearly in dimension. The difference between our work and this work is that we focus on an ultra-sparse sampling regime where there are only $O(n)$ randomly sampled entries, rendering the accurate recovery of the entire matrix information theoretically impossible. In light of our new results, it may also be worth resisting the dynamics of zeroth-order and first-order methods for nonconvex optimization (Ghadimi & Lan, 2013; Sun, 2020) when high rates of noise are added during each iteration.

The idea of using the estimated probability to reweight a population statistic is known as the Hájek estimator (Hájek, 1971). It has often been used to tackle sparse and non-random sampling data (Särndal et al., 2003). This estimator is particularly effective for tackling sparse observational data such as in experimental design (Xiong et al., 2024). It is known from the causal inference literature (Hirano et al., 2003) that the Hájek estimator incurs lower variance than the Horvitz-Thompson estimator, which weights each observation inversely with the true probability. To our knowledge, the analysis of this estimator for one-sided matrix completion appears to be new for the matrix completion literature. Unlike conventional results (Hájek, 1971) where the Hájek estimator is shown to be asymptotically unbiased, for one-sided matrix completion, we find that it is unbiased even in the finite sample regime. Recent work (Bai & Ng, 2021) develops the estimation of counterfactual when potential outcomes follow a factor model of block-missing panel data. Xiong & Pelger (2023) and Duan et al. (2023) develop the inferential theory for latent factor models in large-dimensional panel data with general non-random missing patterns by proposing a PCA-based estimator on an adjusted covariance matrix.

## 7 Conclusion

In this paper, we study the problem of matrix completion in an ultra-sparse sampling regime, where in each row, only a constant number of entries are observed in each row. While recovering the entire matrix is not possible, we focus on the one-sided matrix completion problem. We apply the Hájek estimator from the econometrics literature to this problem, and present a thorough analysis of this estimator. We demonstrate that this estimator is unbiased and achieves a lower variance compared to the Horvitz-Thompson estimator. Then, we use gradient descent to impute the missing entries of the second-moment matrix and analyze its sample complexity under a low-rank mixture model. The sample complexity bound is optimal in its dependence on the dimension $d$. Our result applies to the case of observing $C$ entries at each row, resolving an open question from prior work.

Extensive experiments on both synthetic and real-world datasets demonstrate that our approach is more efficient for sparse matrix datasets compared to several commonly used matrix completion methods. Ablation studies validate the use of a low-rank matrix incoherence regularizer in the algorithm.

Our paper opens up several promising avenues for future work. First, it would be valuable to theoretically analyze the sensitivity of the Hájek estimator to noise injection, which could inform the design of privacy-preserving matrix completion algorithms in the ultra-sparse sampling regime. Second, our current sample-complexity bound exhibits a high dependence on the rank of the latent factor model; improving this rank dependence remains another interesting open question. It would also be interesting to understand to what extent linear sample complexity of $n$ relative to $d$ can be achieved under ultra-sparse sampling, by further relaxing the assumptions required in our analysis. Finally, extending our framework to other forms of panel data such as low-rank tensors or low-rank mixture models, is another promising avenue left for future work.

## Acknowledgment

Thanks to Steven Cao, Daniel Hsu, Oliver Hinder, and Jason Lee for discussions about this work during various stages. We appreciate the editors and the anonymous referees for their constructive comments and suggestions. The work of H. R. Zhang and Z. Zhang is supported in part by NSF award IIS-2412008. The views and opinions expressed in this paper are solely those of the authors and do not necessarily reflect those of the NSF.

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

# A  Proof of Theorem 3.2

First, we present the proof of the unbiasedness of the Hájek estimator on the observed set $\Omega$.

*Proof of Lemma 3.1.* Suppose that $(I^\top I)_{i,j} = k$, for any $k \geq 1$. Then, conditioned on $(I^\top I)_{i,j} = k$, the expectation of $\widehat{T}_{i,j}$ is equal to

$$
\begin{aligned}
\mathbb{E}\left[\widehat{T}_{i,j}\middle|(I^\top I)_{i,j} = k\right] &= \mathbb{E}\left[\frac{1}{k}(\widehat{M}^\top \widehat{M})_{i,j}\middle|(I^\top I)_{i,j} = k\right] \\
&= \mathbb{E}\left[\frac{1}{k}\sum_{a=1}^{n} M_{a,i} M_{a,j} I_{a,i} I_{q,j}\middle|(I^\top I)_{i,j} = k\right] \\
&= \frac{1}{k}\sum_{a=1}^{n} \Pr\left[I_{a,i} I_{a,j} = 1\middle|(I^\top I)_{i,j} = k\right] \cdot M_{a,i} M_{a,j}.
\end{aligned}
$$

Notice that for a fixed $a$, the event that $I_{a,i} I_{a,j} = 1$ are pairwise independent for all $a = 1, 2, \ldots, n$. Given that $k$ out of the $n$ pairs are chosen, the probability that $q$ is chosen is equal to $\binom{n-1}{k-1}/\binom{n}{k} = k/n$. Thus, the above is equal to

$$
\frac{1}{k}\sum_{a=1}^{n} \frac{k}{n} \cdot M_{a,i} M_{a,j} = T_{i,j},
$$

which concludes the proof of equation (3). To finish the proof that $\widehat{T}$ is unbiased, notice that

$$
\begin{aligned}
\mathbb{E}\left[\widehat{T}_{i,j}\middle|\left(I^\top I\right)_{i,j} \neq 0\right] &= \sum_{k=1}^{d} \Pr\left[\left(I^\top I\right)_{i,j} = k\middle|\left(I^\top I\right)_{i,j} \neq 0\right] \cdot \mathbb{E}\left[\widehat{T}_{i,j}\middle|\left(I^\top I\right)_{i,j} = k\right] \\
&= \sum_{k=1}^{d} \Pr\left[\left(I^\top I\right)_{i,j} = k\middle|\left(I^\top I\right)_{i,j} \neq 0\right] \cdot T_{i,j} = T_{i,j},
\end{aligned}
$$

which concludes the proof of equation 2.

Next, we present the proof of Theorem 3.2, which provides a first-order approximation to the variance of the Hájek estimator on $\Omega$.

*Proof of Theorem 3.2.* For equation (4), we use Taylor's expansion on $Var[\widehat{T}_{i,i}]$. For ease of presentation, let us denote $A_{i,i} = (\widehat{M}^\top \widehat{M})_{i,i}$, and let $B_{i,i} = (I^\top I)_{i,i}$, for any $1 \leq i \leq d$. Thus, conditioned on the event that $B_{i,i} \neq 0$, the $(i,i)$-th entry of $\widehat{T}$, given by $\widehat{T}_{i,i}$, is equal to $A_{i,i}/B_{i,i}$.

By Taylor's expansion, we approximate the ratio statistic using the delta method as:

$$
\frac{A_{i,i}}{B_{i,i}} = \frac{\mathbb{E}[A_{i,i}]}{\mathbb{E}[B_{i,i}]} + \frac{1}{\mathbb{E}[B_{i,i}]}(A_{i,i} - \mathbb{E}[A_{i,i}]) - \frac{\mathbb{E}[A_{i,i}]}{(\mathbb{E}[B_{i,i}])^2}(B_{i,i} - \mathbb{E}[B_{i,i}]) + \epsilon_{i,i}, \tag{18}
$$

where $\epsilon_{i,i}$ is at the order of the variance of $A_{i,i}$ and $B_{i,i}$, which is $O(1/(np))$. Thus, based on the first-order expansion, the variance of $\frac{A_{i,i}}{B_{i,i}}$ is approximated by:

$$
Var\left[\frac{A_{i,i}}{B_{i,i}}\right] = \frac{Var[A_{i,i}]}{(\mathbb{E}[B_{i,i}])^2} + \frac{(\mathbb{E}[A_{i,i}])^2 Var[B_{i,i}]}{(\mathbb{E}[B_{i,i}])^4} - \frac{2\mathbb{E}[A_{i,i}] \cdot Cov(A_{i,i}, B_{i,i})}{(\mathbb{E}[B_{i,i}])^3} + O\left(\frac{1}{np}\right), \tag{19}
$$

where the error term is based on the approximation error that stems from $\epsilon_{i,i}$ in equation (18).

Now we look into simplifying equation (19). Based on the definition of $A_{i,i}, B_{i,i}$, we get the following facts, for any $i = 1, \ldots, d$:

$$
\mathbb{E}[B_{i,i}] = np, \quad Var[B_{i,i}] = np(1-p), \tag{20}
$$

$$
\mathbb{E}[A_{i,i}] = p(M^\top M)_{i,i}, \quad Cov[A_{i,i}, B_{i,i}] = (p - p^2)(M^\top M)_{i,i}. \tag{21}
$$

To verify the above results, let us introduce a Bernoulli random variable $X_k$, which is equal to 1 with probability $p$, or 0 with probability $1 - p$, for $k = 1, 2, \ldots, n$. Thus, we can write

$$A_{i,i} = \sum_{k=1}^{n} M_{k,i}^2 X_k, \text{ and } B_{i,i} = \sum_{k=1}^{n} X_k.$$

As a result,

$$
\begin{aligned}
Cov[A_{i,i}, B_{i,i}] &= \mathbb{E}[A_{i,i} B_{i,i}] - \mathbb{E}[A_{i,i}] \cdot \mathbb{E}[B_{i,i}] \\
&= \mathbb{E}\left[\left(\sum_{k=1}^{n} M_{k,i}^2 X_k\right)\left(\sum_{k=1}^{n} X_k\right)\right] - np^2 (M^\top M)_{i,i} \\
&= \mathbb{E}\left[\left(\sum_{k=1}^{n} M_{k,i}^2 X_k\right) + \left(\sum_{1 \le k \ne k' \le n} M_{k,i}^2 X_k X_{k'}\right)\right] - np^2 (M^\top M)_{i,i} \quad (\text{note that } X_k^2 = X_k) \\
&= \left(\sum_{k=1}^{n} M_{k,i}^2\right)(p + (n-1)p^2) - np^2 (M^\top M)_{i,i} \\
&= (p - p^2)(M^\top M)_{i,i}.
\end{aligned}
$$

Therefore, by plugging in the results from equations (20) and (21) back into equation (19), we obtain an approximate variance of the ratio statistic as:

$$
\begin{aligned}
&\frac{Var[A_{i,i}]}{n^2 p^2} + \frac{p^2 (M^\top M)_{i,i}^2 \cdot np(1-p)}{n^4 p^4} - \frac{2p(M^\top M)_{i,i} \cdot (p - p^2)(M^\top M)_{i,i}}{n^3 p^3} \\
&= \frac{Var[A_{i,i}]}{n^2 p^4} - \frac{(1-p)(M^\top M)_{i,i}^2}{n^3 p}.
\end{aligned}
\tag{22}
$$

This applies to any diagonal entries of $\widehat{T}$. In particular, notice that the first term of equation (22) is precisely the variance of $\overline{T}_{i,i}$, which reweights the observed entries based on the true probability. This leads to the following variance estimate for the diagonal entries:

$$\frac{Var[A_{i,i}]}{n^2 p^2} - \frac{(1-p)(M^\top M)_{i,i}^2}{n^3 p}. \tag{23}$$

Finally, we write down the variance of $A_{i,i}$ as follows:

$$Var[A_{i,i}] = p(1-p) \sum_{k=1}^{n} M_{k,i}^4. \tag{24}$$

Notice that the probability that $B_{i,i}$ is equal to zero is at most $O(d^{-C})$. We can add this error analysis into the above proof, which does not affect the order of the error term. This concludes the proof of equation (4) in Theorem 3.2 regarding the variance approximation of the diagonal entries.

Next, we consider the case of off-diagonal entries in equation (5). The probability that an off-diagonal entry of $\widehat{T}$ is nonzero is

$$1 - (1 - p^2)^n,$$

which is approximately $np^2 = C^2 (\log d)^2 / d$. Thus, conditioned on the fact that $(I^\top I)_{i,j} \ne 0$, the dominating event is when $(I^\top I)_{i,j} = 1$. The error to this, which is when there are at least two nonzero entries in the sum, is less than

$$\frac{\Pr[(I^\top I)_{i,j} \ge 2]}{\Pr[(I^\top I)_{i,j} \ne 0]} \approx np^2.$$

When the $(i,j)$-th entry of $I^\top I$ is equal to one, then $\widehat{T}_{i,j} = (\widehat{M}^\top \widehat{M})_{i,j}$. The variance of $\widehat{T}_{i,j}$, conditioned on $(I^\top I)_{i,j} = 1$, is thus given by

$$\frac{1}{n} \sum_{k=1}^{n} \left(M_{k,i} M_{k,j} - \frac{1}{n}\left(\sum_{k'=1}^{n} M_{k',i} M_{k',j}\right)\right)^2.$$

After simplifying the above formula, we reach the conclusion as stated in equation (5). The proof is thus completed.

**Remark A.1** (Potential applications of the Hájek estimator)**.** *Beyond one-sided matrix completion, Hájek estimators may serve as a useful tool in other high-dimensional estimation problems given few observational samples. In tensor completion (Zhang et al., 2019a), naive inverse-propensity weighting estimators can exhibit large variance due to rare observations, whereas self-normalization can stabilize the estimation of low-rank factors given ultra-sparse samples.*

*Similarly, in off-policy reinforcement learning (Zhang et al., 2025), importance weighted estimators are widely used to correct for distribution mismatch but are often unstable as in sparse-reward settings. Hájek-style estimators provide a principled variance-reduction mechanism and may be useful for estimating value-function matrices under partial trajectory observations.*

## B  Proof of Theorem 3.4

We analyze the sample complexity of Algorithm 1. In particular, we will analyze the loss function $\ell(X)$, including the incoherence regularization penalty $R(X)$. See equation (8). Without loss of generality, let us assume that $\|U\|_F^2 = r$ within this section. This also implies $\sigma_{\max}(U) \geq 1 \geq \sigma_{\min}(U)$.

A key part of the proof is analyzing the bias of $\widehat{T}$ (relative to $T$). Let $N \in \mathbb{R}^{d \times d}$ denote the bias of the Hájek estimator, where

$$N_{i,j} = \widehat{T}_{i,j} - T_{i,j}, \quad \forall (i,j) \in \Omega,$$
$$N_{i,j} = 0, \quad \forall (i,j) \notin \Omega.$$

Here is a road map of this section:

- First, in Appendix B.1, we derive upper bounds on the variance of the Hájek estimator, on the observed set $\Omega$.

- Second, in Appendix B.2, we derive a concentration bound that applies to the weighted projection $P_\Omega$. By a similar argument, we also derive the spectral norm of the noise matrix $N$.

- Third, in Appendix B.3, we derived the local optimality conditions from analyzing the loss function $\ell(\cdot)$, and then we argue that any local minimum solution of $\ell(\cdot)$ must satisfy the incoherence assumption with a suitable condition number.

- Fourth, in Appendix B.4, we use the incoherence assumption to reduce the local optimality conditions on the empirical loss to the population loss.

Finally, in Appendix B.5, we analyze the population loss and based on that, complete the proof of Theorem 3.4. The key is to use the regularization penalty $R(X)$ to control the norm of each factor, leading to concentration estimates between the population loss and the empirical loss. We provide a detailed comparison between this work and the work of Ge et al. (2016) in Remark B.12.

### B.1  Proof of Lemma 4.1

In the following proof, we proceed to bound the variance of the diagonal entries of $\widehat{T}$ and also its off-diagonal entries grouped by each row.

*Proof.* Based on Lemma 3.1, we already know that $\widehat{T}$ is unbiased on $\Omega$. Next, we derive the variance of the diagonal and off-diagonal entries of $\widehat{T}$ separately, since

$$\mathbb{E}\left[(\widehat{T}_{i,j} - T_{i,j})^2\right] = Var[\widehat{T}_{i,j}], \text{ for any } (i,j) \in \Omega. \tag{25}$$

For the diagonal entries of $T$, with probability at least $1 - O(d^{-1})$, all of them are observed in $\Omega$. In particular, when $n \geq d \log d / \epsilon^2$, by the Chernoff bound and union bound, with probability at least $1 - O(d^{-1})$, for all $i = 1, 2, \ldots, d$,

$$\left| (I^\top I)_{i,i} - \mathbb{E}\left[ (I^\top I)_{i,i} \right] \right| \leq \epsilon \sqrt{\frac{6C^{-1}}{d} np}.$$

Thus, we can get that

$$Var[\widehat{T}_{i,i}] \leq \left( 1 + 3\epsilon \sqrt{\frac{6C^{-1}}{d}} \right) Var\left[ \frac{A_{i,i}}{np} \right].$$

Next, notice that

$$A_{i,i} = \sum_{k=1}^{n} M_{k,i}^2 X_k. \tag{26}$$

Therefore, the variance of $\frac{A_{i,i}}{np}$ is equal to

$$
\begin{aligned}
Var\left[ \frac{A_{i,i}}{np} \right] &= \frac{1-p}{n^2 p} \sum_{k=1}^{n} M_{k,i}^4 \\
&\leq \frac{1-p}{np} \left( \frac{1}{r} \sum_{s=1}^{r} u_{i,s}^4 \right) \left( 1 + \frac{\max_{1 \leq s \leq r} u_{i,s}^4}{r^{-1} \sum_{s=1}^{r} u_{i,s}^4} \sqrt{\frac{3 \log(d/2)}{2n}} \right) \quad \text{(by Hoeffding's inequality)} \\
&\leq \frac{1-p}{np} \left( \frac{1}{r} \left( \sum_{s=1}^{r} u_{i,s}^2 \right)^2 \right) \left( 1 + r \sqrt{\frac{3 \log(d/2)}{2n}} \right) \tag{27} \\
&\leq \frac{1-p}{np} \frac{\mu^2 r}{d^2} \left( 1 + r \sqrt{\frac{3 \log(d/2)}{2n}} \right),
\end{aligned}
$$

which concludes the proof of equation (11). Similar to this calculation, by applying Hoeffding's inequality, we can also show that

$$
\begin{aligned}
\frac{A_{i,i}}{np} &\leq \frac{1}{r} \sum_{s=1}^{r} \mu_{i,s}^2 \left( 1 + r \sqrt{\frac{3 \log(d/2)}{2n}} \right) \\
&\leq \frac{\mu}{d} \left( 1 + r \sqrt{\frac{3 \log(d/2)}{2n}} \right). \tag{28}
\end{aligned}
$$

Next, we examine the off-diagonal entries of $\widehat{T}$. From Theorem 3.2, the variance of the off-diagonal entries is approximated by the variance of a single entry. Consider the case where we condition on taking two overlapping entries from columns $i$ and $j$. Denote the two entries as $a_1, a_2$. We have that

$$
\begin{aligned}
Var\left[ \frac{a_1 + a_2}{2} \right] &= \frac{Var[a_1] + Var[a_2]}{4} + Cov[a_1, a_2] \\
&\leq \frac{Var[a_1] + Var[a_2]}{2},
\end{aligned}
$$

by the Cauchy-Schwarz inequality on the covariance. The probability that we observe three overlapping entries is less than $\tilde{O}(d^{-3})$. Since the size of $\Omega$ is $O(d^2 q) = \tilde{O}(d^{-1})$. The probability that such events can happen is less than $\tilde{O}(d^{-1})$. In summary, we have shown that for every $(i, j) \in \Omega$ where $i \neq j$,

$$
\begin{aligned}
Var[\widehat{T}_{i,j}] &\leq \left( \frac{1}{n} \left( \sum_{k=1}^{n} M_{k,i}^2 M_{k,j}^2 \right) - T_{i,j}^2 \right) + \tilde{O}(d^{-3}) \\
&\leq \frac{1}{n} \sum_{k=1}^{n} M_{k,i}^2 M_{k,j}^2 + \tilde{O}(d^{-3}).
\end{aligned}
$$

We now look at the sum of the variances of all the observed entries in the $i$-th row. Let the set be denoted by $S_i = \{j : (i,j) \in \Omega, j \neq i\}$, for all $i = 1, 2, \ldots, d$. We have

$$\frac{1}{n} \sum_{j \in S_i} \sum_{k=1}^{n} M_{k,i}^2 M_{k,j}^2 \tag{29}$$

$$\leq \sum_{j \in S_i} \left( \frac{1}{r} \sum_{s=1}^{r} u_{i,s}^2 u_{j,s}^2 \right) \left( 1 + \frac{\max_{s=1}^{r} u_{i,s}^2 u_{j,s}^2}{r^{-1} \sum_{s=1}^{r} u_{i,s}^2 u_{j,s}^2} \sqrt{\frac{3 \log(d/2)}{2n}} \right) \qquad \text{(by Hoeffding's inequality)}$$

$$\leq \frac{1}{r} \sum_{s=1}^{r} u_{i,s}^2 \cdot q \left\| u_s \right\|^2 \left( 1 + \frac{\max_{j=1}^{d} u_{j,s}^2}{r^{-1} \sum_{j=1}^{d} u_{j,s}^2} \sqrt{\frac{3 \log(d/2)}{2d}} \right) \left( 1 + r \sqrt{\frac{3 \log(d/2)}{2n}} \right) \qquad \text{(by Hoeffding's inequality)}$$

$$\leq \frac{q}{r} \cdot \max_{1 \leq s' \leq r} \left\| u_{s'} \right\|^2 \cdot \sum_{s=1}^{r} u_{i,s}^2 \cdot \left( 1 + r \sqrt{\frac{3 \log(d/2)}{2d}} \right) \left( 1 + r \sqrt{\frac{3 \log(d/2)}{2n}} \right) \qquad \text{(relax } \left\| u_s \right\|^2 \text{ by its max over } s)$$

$$\leq \frac{\mu q}{d} \cdot \left( 1 + 3r \sqrt{\frac{3 \log(d/2)}{2d}} \right) \qquad \text{(by the incoherence assumption on } U)$$

The last line uses the premise that the maximum $\ell_2$ norm of a factor is at most 1, and also $n \geq d$. In particular, the above Hoeffding's inequality applies uniformly to all $i$ with probability at least $1 - O(d^{-1})$. In summary, we have shown that

$$\sum_{j \in S_i} Var[\widehat{T}_{i,j}] \leq \frac{\mu q}{d} \left( 1 + 3r \sqrt{\frac{3 \log(d/2)}{2d}} \right) + \tilde{O}\left( \frac{nq}{d^3} \right),$$

where we use the fact that with probability at least $1 - d^{-2}$, $|S_i| \leq 2nq$. Therefore, we conclude that equation (12) is true.

As a remark, one corollary we can draw from the above calculation is that

$$Var[\widehat{T}_{i,j}] \leq \frac{\mu^2 r^2}{d^2} \left( 1 + r \sqrt{\frac{3 \log(d/2)}{2n}} \right) + \tilde{O}(d^{-3}). \tag{30}$$

This can be seen by following the steps from equation (29) for a fixed pair of $(i,j) \in \Omega$.

**Remark B.1.** *In the case that $M_i$ is a noisy perturbation of one of the $r$ factors, we can extend the proof following equation (29). Another natural extension is to assume that every $M_i$ is a linear combination of the $r$ factors, $\mu_1, \ldots, \mu_r$. This seems to be a much more challenging problem, as one would need to design estimators to first disentangle the linear combination patterns. This is left for future work.*

Next, we build on Lemma 3.1 to examine the bias of $\widehat{T}$ on the observed set $\Omega$. Another corollary from the above variance calculation is the following bound on the bias of $\widehat{T}$. We consider the diagonal and off-diagonal entries separately.

**Corollary B.2.** *In the setting of Lemma 4.1, suppose $n \geq dr$. Then, with probability at least $1 - O(d^{-1})$, for every $i = 1, 2, \ldots, d$, the following must hold:*

$$N_{i,i} \leq \sqrt{\frac{32 C^{-1} \mu^2 \log(d)}{nd^2}}, \tag{31}$$

$$\sum_{j \in S_i} |N_{i,j}| \leq q \cdot \sqrt{2\mu + \tilde{O}(d^{-1/2})}. \tag{32}$$

*Proof.* First, we have that

$$|N_{i,i}| = \left| \widehat{T}_{i,i} - T_{i,i} \right| = \left| \frac{A_{i,i}}{B_{i,i}} - T_{i,i} \right| = \left| \left( 1 \pm \epsilon \sqrt{\frac{6C^{-1}}{d}} \right) \frac{A_{i,i}}{np} - T_{i,i} \right|$$
$$\leq \left| \frac{A_{i,i}}{np} - T_{i,i} \right| + \epsilon \sqrt{\frac{6C^{-1}}{d}} \frac{A_{i,i}}{np}.$$

By equation (28), the latter is at most

$$\epsilon \sqrt{\frac{6C^{-1}}{d}} \frac{\mu}{d} \left( 1 + r \sqrt{\frac{3 \log(d/2)}{2n}} \right) \leq \sqrt{\frac{8C^{-1}\mu^2 \log(d)}{d^2 n}},$$

since $\epsilon \leq \sqrt{d \log d / n}$. As for the former, by equation (31), we have

$$Var[N_{i,i}] \leq (1 + \epsilon \sqrt{6C^{-1}/d}) Var\left[ \frac{\widehat{A}_{i,i}}{np} \right].$$

Recall that $A_{i,i}$ is the sum of $n$ Bernoulli random variables (cf. equation (26)), by Bernstein's inequality, with probability at least $1 - O(d^{-1})$, for all $i = 1, 2, \ldots, d$,

$$\left| \frac{A_{i,i}}{np} - T_{i,i} \right| \leq \sqrt{\frac{6Var(N_{i,i}) \cdot \log(d/2)}{n}} + \frac{3 \left( \max_{s=1}^{r} u_{i,s}^2 \right) \log(d/2)}{3n}$$
$$\leq \sqrt{\frac{7\mu^2 r \log(d/2)}{d^2 n^2 p}} + \frac{\mu^2 r^2 \log(d/2)}{nd},$$

for large enough values of $d$. Specifically, we have used the bound on the variance of $\widehat{T}_{i,i}$ in Lemma 4.1, using equation (11). Notice that the second term is a lower-order term relative to the first. This shows that equation (31) holds for large enough values of $d$.

For equation (32), based on the proof of Lemma 4.1,

$$\sum_{j \in S_i} |N_{i,j}| \leq \sqrt{|S_i| \cdot \sum_{j \in S_i} N_{i,j}^2} \qquad \text{(by the Cauchy-Schwarz inequality)}$$
$$\leq \sqrt{2dq \cdot \sum_{j \in S_i} N_{i,j}^2} \qquad \text{(since } |S_i| \leq 2dq)$$
$$\leq \sqrt{2dq \cdot \left( \frac{\mu q}{d} \left( 1 + 3r \sqrt{\frac{3 \log(d/2)}{2d}} \right) + \tilde{O}(d^{-3}) + \frac{2\mu^2 r^2}{d^2} \cdot \sqrt{\frac{3 \log(d/2)}{2d}} \right)}, \qquad (33)$$

where the last line is by applying Hoeffding's inequality to the sequence $\left\{ N_{i,j}^2 : j \in S_i \right\}$ (which are all independent from each other), and holds with probability at least $1 - d^{-1}$ over all possible $i$. In particular, the expectation on this sequence is based on equation (12) and the maximum is based on equation (30). From the last line above, we conclude that equation (32) is true.

## B.2    Proof of Lemma 4.2

*Proof.* By the weighted operation in $P_{\Omega}$ and the condition that all the diagonals have been observed, we can cancel out the diagonal entries between $P_{\Omega}(W)$ and $qW$. In the rest of the following, we focus on the off-diagonal entries. Let $x \in \mathbb{R}^d$ denote a unit vector. We have that

$$\|P_{\Omega}(W) - qW\|_2 = \max_{x: \|x\|=1} x^\top (P_{\Omega}(W) - qW)x.$$

Next, we expand the right-hand side above as:

$$x^\top (P_\Omega(W) - qW)x = \sum_{i=1}^d \left( \sum_{j \in S_i : j \neq i} W_{i,j} x_i x_j - q \sum_{1 \leq j \leq d : j \neq i} W_{i,j} x_i x_j \right)$$

$$\leq \sum_{i=1}^d \underbrace{\left| \sum_{j \in S_i : j \neq i} W_{i,j} x_i x_j - q \sum_{1 \leq j \leq d : j \neq i} W_{i,j} x_i x_j \right|}_{e_i}.$$

We focus on a fixed $i$ above. By Bernstein's inequality, with probability at least $1 - O(d^{-2})$, we have

$$e_i \leq \sqrt{4q(1-q) \sum_{1 \leq j \leq d : j \neq i} \left( W_{i,j}^2 x_i^2 x_j^2 \right) \log(2d)} + \|W\|_\infty |x_i| \max_{1 \leq j \leq d} |x_j|$$

$$\leq \|W\|_\infty |x_i| \sqrt{4q(1-q) \log(2d)} + \|W\|_\infty |x_i| \max_{1 \leq j \leq d} |x_j|.$$

Above, we have plugged in the variance and the maximum values at the $i$-th row. As a result,

$$\sum_{i=1}^d e_i \leq \frac{\nu}{\sqrt{d}} \sqrt{4q(1-q) \log(2d)} + \frac{\nu}{d} \sum_{i=1}^d |x_i| \max_{1 \leq j \leq d} |x_j|. \tag{34}$$

In particular, we have used the bound on $\|W\|_\infty \leq \nu/\sqrt{d}$ and the fact that $\sum_i |x_i| \leq \sqrt{d}$ via the Cauchy-Schwarz inequality.

Next, we notice that

$$\sum_{i=1}^d |x_i| \max_{1 \leq j \leq d} |x_j| \leq 1.$$

To see this, notice that if we look at a particular pair $x_i, x_j$. Conditioned on a fixed $x_i^2 + x_j^2$ and $|x_i| \leq \lambda$, $|x_j| \leq \lambda$, $|x_i| + |x_j|$ are maximized when $x_i = x_j$. This implies that when $\sum_{i=1}^d |x_i|$ is maximized, all the non-zero $x_i$'s must be equal to each other. Suppose there are $k$ of them, and they are all equal to $a$. Then, we must have $ka^2 \leq 1$, $a \leq \lambda$. As a result,

$$\sum_{i=1}^d |x_i| \cdot \max_j |x_j| = ka\lambda \leq 1.$$

By rearranging equation (34), and using the fact that $dq \geq 4\log(2d)$ to relax the second term in equation (34), we have concluded the proof of equation (15).

With a similar proof, we can derive a bound on the spectral norm of $N$, stated as follows.

**Corollary B.3.** *In the setting of Lemma 4.1, as long as $q \geq \frac{4\log(2d)}{d}$, then with probability at least $1 - O(d^{-1})$, the spectral norm of $P_\Omega(N)$ satisfies:*

$$\|P_\Omega(N)\|_2 \leq q\sqrt{\frac{32\mu^2 r^2 \log(2d)}{dq}} + O\left( q\sqrt{\frac{\log(d)}{nd^2}} \right). \tag{35}$$

*Proof.* Let $x \in \mathbb{R}^d$ be any unit vector. Note that $N$ is a symmetric matrix. Therefore, we expand $x^\top N x$ as follows:

$$x^\top N x = q \underbrace{\sum_{i=1}^d N_{i,i} x_i^2}_{e_1} + \underbrace{\sum_{1 \leq i \leq d} \sum_{j \in S_i : j \neq i} N_{i,j} x_i x_j}_{e_2}.$$

By equation (31), Corollary B.2,

$$e_1 \leq q\sqrt{\frac{32C^{-1}\mu^2\log(d)}{nd^2}\sum_{i=1}^{d}x_i^2} = q\sqrt{\frac{32C^{-1}\mu^2\log(d)}{nd^2}}.$$

As for $e_2$, notice that $|N_{i,j}| = \left|\widehat{T}_{i,j} - T_{i,j}\right| \leq 2\mu r/d$. By following the proof of Lemma 4.2 above (cf. equation (34)), we have that

$$e_2 \leq \frac{2\mu r}{\sqrt{d}}\sqrt{4q(1-q)\log(2d)} + \frac{2\mu r}{d}.$$

Since $dq \geq 4\log(2d)$, the second term on the right is less than or equal to the first term on the right. Put together, we can see that equation (35) is true.

Another implication is the inner product of two matrices under the projection operator.

**Proposition B.4.** *Suppose* $\|X\|_\infty \cdot \|Y\|_\infty \leq \frac{\nu}{d}$. *With probability at least* $1 - O(d^{-1})$ *over the randomness of* $\Omega$, *for any two matrices* $X, Y \in \mathbb{R}^{d \times d}$, *we have*

$$|\langle P_\Omega(X), Y\rangle - q\langle X, Y\rangle| \leq q\nu\sqrt{\frac{16\log(2d)}{dq}}. \tag{36}$$

*Proof.* With probability at least $1 - O(d^{-1})$, the diagonal entries of $\langle P_\Omega(X), Y\rangle$ are canceled out with that of $q\langle X, Y\rangle$. As for the off-diagonal entries, let

$$e_2 = \sum_{1\leq i\leq d}\sum_{j\in S_i: j\neq i} X_{i,j}Y_{i,j} - q\sum_{1\leq i\leq d}\sum_{1\leq j\leq d: j\neq i} X_{i,j}Y_{i,j}.$$

Notice that for a fixed $i$, whether $j \in S_i$ or another $j' \in S_i$ are independent events. Thus, we apply Bernstein's inequality to each row, similar to the steps following equation (34), which leads to the bound on $e_2$ stated in equation (36).

### B.3 Local Optimality and Incoherence

First, we derive the first-order and second-order optimality conditions of $\ell(X)$, accounting for the weighted projection $P_\Omega$ to offset the imbalance between diagonal and off-diagonal entries in $\Omega$.

**Proposition B.5** (See also Proposition 5.1, Ge et al. (2016))**.** *Suppose* $X$ *is a local optimum of* $\ell(X)$ *(See equation* (8)*). Then, the first-order optimality condition is equivalent to*

$$2P_\Omega(\widehat{T})X = 2P_\Omega(XX^\top)X + \lambda\nabla R(X). \tag{37}$$

*The second-order optimality condition is equivalent to:*

$$\forall V \in \mathbb{R}^d, \quad \langle P_\Omega(VX^\top + XV^\top), XV^\top + VX^\top\rangle + \lambda\langle V, \nabla^2 R(X)V\rangle$$
$$\geq 2\langle P_\Omega(\widehat{T} - XX^\top), VV^\top\rangle. \tag{38}$$

*Proof.* The gradient formula of equation (37) can be seen by directly taking the gradient on $\ell(X)$, and separating the gradient from the squared loss vs the regularizer. As for the Hessian formula of equation (38), when $V$ is sufficiently small, one has that $\nabla\ell(X+V) = [\nabla^2\ell(X)]V$. Thus, we have that

$$\nabla\ell(X+V)$$
$$= 2P_\Omega(VX^\top + XV^\top + VV^\top + XX^\top)(X+V) - 2P_\Omega(\widehat{T})(X+V) + \lambda\nabla R(X+V).$$

By the second-order optimality condition, one has that $V^\top \nabla \ell(X + V) \geq 0$, when $V$ tends to zero. Plugging this into the above, we get

$$V^\top \nabla \ell(X + V)$$
$$= 2\langle P_\Omega(VX^\top + XV^\top), XV \rangle + 2\langle P_\Omega(XX^\top), VV^\top \rangle - 2\langle P_\Omega(\widehat{T}), VV^\top \rangle + \lambda \nabla R(X + V) + e$$
$$= \langle P_\Omega(VX^\top + XV^\top), XV^\top + VX^\top \rangle + 2\langle P_\Omega(XX^\top - \widehat{T}), VV^\top \rangle + \lambda \langle V, \nabla^2 R(X)V \rangle + e$$
$$= \langle P_\Omega(VX^\top + XV^\top), VX^\top + XV^\top \rangle + 2\langle P_\Omega(XX^\top - \widehat{T}), VV^\top \rangle + \lambda \langle V, \nabla^2 R(X)V \rangle + e,$$

where $e = O(\|V\|_F^2)$ denotes an error term. When $V$ tends to zero, we have that equation (38) must be true for any $V \in \mathbb{R}^d$ based on the second-order optimality condition.

Next, we show that the regularizer of $R(X)$ leads the row norms of $X$ to be bounded by at most $2\alpha$, which builds on the first-order optimality conditions above. As a result, we can use concentration tools to reduce the first- and second-order optimality conditions for their population versions. Based on that, we show that $X$ must be bounded away from zero.

We begin by analyzing the effect of the regularizer. We show the following property, which results from adding the incoherence regularizer and can be derived based on the gradient of the regularizer.

**Proposition B.6.** *The gradient of $R(X)$ satisfies that for any $Y \in \mathbb{R}^{d \times r}$,*

$$\langle \nabla R(X), Y \rangle = 4\lambda \sum_{i=1}^d \left( \|X_i\|^3 - \alpha \right)_+^3 \frac{\langle X_i, Y^\top e_i \rangle}{\|X_i\|}.$$

*where $e_i \in \mathbb{R}^d$ is the i-th basis vector. As a consequence,*

$$\langle (\nabla R(X))_i, X_i \rangle \geq 0, \text{ for every } i = 1, 2, \dots, d.$$

We first notice that because of the regularizer, any matrix $X$ that satisfies the first-order optimality condition must also satisfy the incoherence condition.

**Lemma B.7.** *Let $S_i$ denote the set of observed entries in $\widehat{I}$ at the i-th row, for $i = 1, 2, \dots, d$. Suppose $|S_i| \leq 2dq$, for any $i = 1, 2, \dots, d$. In the setting of Theorem 3.4, for any $X$ that satisfies the first-order optimality of equation (37), we have*

$$\max_{i=1}^d \|X_i\| \leq \max\left( 2\alpha, 2\sqrt{\mu(r+1)q/\lambda} \right). \tag{39}$$

*Proof.* Let $i^\star = \arg\max_{1 \leq i \leq d} \|X_i\|$ be the index of the row that has the highest norm in $X$. Recall that $X_i \in \mathbb{R}^r$ refers to the i-th row vector of $X$. Suppose the i-th row of $\Omega$ consists of entries with index $[i] \times S_i$, where $S_i$ is the set of indices in the i-th row that are observed in $\Omega$. If $|X_{i^\star}| \leq 2\alpha$, then equation (39) is true. Otherwise, let us assume that $|X_{i^\star}| \geq 2\alpha$.

We will compare the $i^\star$-th row of the left and right-hand sides of equation (37). First, we have the following identity based on the projection matrix $P_\Omega$:

$$\left( P_\Omega(\widehat{T})X \right)_{i^\star} = \left( P_\Omega(UU^\top + N)X \right)_{i^\star} = \left( P_\Omega(UU^\top) \right)_{i^\star} X + N_{i^\star} X,$$

where the sub-index on $i^\star$ refers to the $i^\star$-th row of the enclosed matrix. Then, the $\ell_1$-norm of $P_\Omega(UU^\top)$ is at most

$$\left\| \left( P_\Omega(UU^\top) \right)_{i^\star} \right\|_1 = \sum_{j \in S_{i^\star}} |\langle U_{i^\star}, U_j \rangle| \tag{40}$$

$$\leq \sum_{j \in S_{i^\star}} \|U_{i^\star}\| \cdot \|U_j\| \leq \sum_{j \in S_{i^\star}} \frac{\mu r}{d} \qquad \text{(by the incoherence assumption on } U\text{)}$$

$$\leq \frac{2\mu r d q}{d} = 2\mu r q. \qquad \text{(by } |S_{i^\star}| \leq 2dq\text{)}$$

Second, we look at the $\ell_1$-norm of the $i^\star$-th row of $N$. By using Corollary B.2, we can bound the $\ell_2$-norm of the left of equation (37) by

$$
\begin{aligned}
\left\|(P_\Omega(\widehat{T})X)_{i^\star}\right\| &\leq \left(\left\|(P_\Omega(UU^\top))_{i^\star}\right\|_1 + \left\|(P_\Omega(N))_{i^\star}\right\|_1\right) \cdot \max_{i=1}^{d}\|X_i\| \\
&\leq \left(2\mu r + \sqrt{2\mu + \tilde{O}(d^{-1/2})} + \tilde{O}(n^{-1})\right) q \left\|X_{i^\star}\right\|,
\end{aligned}
\tag{41}
$$

where we use equations (40), (31), and (32) above, along with the fact that $p = \frac{C}{n}$.

Next, we lower bound the norm of the right-hand side of equation (37). We have that

$$
(P_\Omega(XX^\top)X)_{i^\star} = \sum_{j \in S_{i^\star}} \langle X_{i^\star}, X_j\rangle X_j.
$$

Thus, by taking an inner product with $X_{i^\star}$, we get

$$
\langle (P_\Omega(XX^\top)X)_{i^\star}, X_{i^\star}\rangle = \sum_{j \in S_{i^\star}} \langle X_{i^\star}, X_j\rangle^2 \geq 0.
$$

Using Proposition B.6, we obtain that

$$
\langle (P_\Omega(XX^\top)X)_{i^\star}, (\nabla R(X))_{i^\star}\rangle = 4\lambda \left(|X_{i^\star}| - \alpha\right)_+^3 \frac{\sum_{j \in S_{i^\star}} \langle X_{i^\star}, X_j\rangle^2}{\|X_{i^\star}\|} \geq 0.
\tag{42}
$$

It follows that

$$
\begin{aligned}
\left\|(P_\Omega(XX^\top)X)_{i^\star} + (\lambda\nabla R(X))_{i^\star}\right\| &\geq \left\|(\lambda\nabla R(X))_{i^\star}\right\| && \text{(by equation (42))} \\
&= \frac{4\lambda(\|X_{i^\star}\| - \alpha)_+^3}{\|X_{i^\star}\|} \cdot \|X_{i^\star}\| && \text{(by Proposition B.6)} \\
&\geq \frac{\lambda}{2} \|X_{i^\star}\|^3 && \text{(since } \|X_{i^\star}\| \geq 2\alpha)
\end{aligned}
$$

Therefore, plugging in the equation above and the equation (41) into the first-order optimality condition (37). We obtain equation (39). Thus, the proof is completed.

The condition that $|S_i| \leq 2dq$ for all $i$ can be shown via Chernoff bounds (recall that $q = 1 - (1 - p^2)^n$). To see this, notice that conditioned on row $i$, the event of whether $j \in S_i$ (for any $j$ between 1 and $d$ but not equal to $i$) is independent from each other. In expectation, the size of $|S_i|$ should be equal to $dq$. In the setting of Theorem 3.4, $dq$ is at least $2\log(d)$. Therefore, with probability at least $1 - O(d^{-2})$, $|S_i| \leq 2dq$, for all $i = 1, 2, \ldots, d$.

### B.4 Reduction of Local Optimality Conditions

One consequence of the above result is the following population version of the first-order optimality condition.

**Lemma B.8.** *Suppose $\alpha \leq \sqrt{\mu(r+1)q/\lambda}$ and $\lambda \geq dq/8$. In the setting of Theorem 3.4, with high probability over the randomness of $\Omega$, for any $X \in \mathbb{R}^{d \times r}$ that satisfies the first-order optimality condition (37), we have that*

$$
\|X\|_F \leq \sigma_{\max}(U)\sqrt{r} + \sqrt{\delta}, \ \text{where } \delta = 3\sqrt{32\mu^2 r^3 \log(2d)/(dq)},
\tag{43}
$$

$$
\left\|UU^\top X - XX^\top X - \gamma\nabla R(X)\right\|_F \leq 42\sigma_{\max}(U)\sqrt{\frac{\mu^2 r^4 \log(2d)}{dq}}.
\tag{44}
$$

*Proof.* By the incoherence assumption on $U$, we have that

$$
\left\|UU^\top\right\|_\infty \leq \frac{\mu r}{d}\|U\|_F^2 \leq \frac{\mu r^{3/2}}{d}\left\|UU^\top\right\|_F,
$$

by applying the Cauchy-Schwarz inequality. By setting $\nu = \mu r^{3/2} \left\| UU^\top \right\|_F$ in Lemma 4.2, from equation (15) we get

$$
\begin{aligned}
\left\| P_\Omega(UU^\top)X - qUU^\top X \right\|_F &\leq \left\| P_\Omega(UU^\top)X - qUU^\top \right\|_2 \cdot \|X\|_F \\
&\leq q\delta_1 \|X\|_F,
\end{aligned}
\tag{45}
$$

for $\delta_1 = 4\sqrt{\mu^2 r^3 \log(2d)/(dq)}$.

Next, by the incoherence condition on $X$ in equation (39), we have that

$$
\|X\|_\infty \leq \frac{4\mu(r+1)q}{\lambda}.
$$

Using equation (15) from Lemma 4.2 with $\nu = 4\mu(r+1)dq/\lambda$, we have

$$
\begin{aligned}
\left\| P_\Omega(XX^\top)X - qXX^\top X \right\|_F &\leq \left\| P_\Omega(XX^\top) - qXX^\top \right\|_2 \|X\|_F \\
&\leq q\delta_2 \|X\|_F,
\end{aligned}
\tag{46}
$$

where

$$
\delta_2 = 4\mu(r+1)\sqrt{dq\log(2d)}/\lambda \leq 32\mu(r+1)\sqrt{\log(2d)/(dq)},
$$

for $\lambda \geq dq/8$.

We also need to add the error induced by $N$, given by

$$
\begin{aligned}
\|P_\Omega(N)X\|_F &\leq \|P_\Omega(N)\|_2 \cdot \|X\|_F \\
&\leq q\delta_3 \|X\|_F,
\end{aligned}
\tag{47}
$$

where $\delta_3 = \sqrt{32\mu^2 r^2 \log(2d)/(dq)} + \tilde{O}(n^{-1}d^{-2})$, and the bound on the spectral norm of $P_\Omega(N)$ is shown in Corollary B.3.

Now, by plugging equations (45), (46), and (47) into the first-order optimality condition (cf. equation (37)), we have shown that

$$
\left\| UU^\top X - XX^\top X - \gamma \nabla R(X) \right\|_F \leq \delta \|X\|_F,
\tag{48}
$$

where $\delta = \delta_1 + \delta_2 + \delta_3$.

When $\|X\|_F \leq \sigma_{\max}(U)\sqrt{r}$, both equations (43) and (44) are proved. Otherwise, by the triangle inequality

$$
\begin{aligned}
q\left\| UU^\top X \right\|_F &\geq \left\| P_\Omega(UU^\top)X \right\|_F - q\delta_1 \|X\|_F \\
&= \left\| P_\Omega(XX^\top)X + \lambda R(X) \right\|_F - q\delta_1 \|X\|_F - \|P_\Omega(N)\|_2 \|X\|_F \\
&\geq \left\| P_\Omega(XX^\top)X \right\|_F - q(\delta_1 + \delta_3) \|X\|_F \\
&\geq q\left\| XX^\top X \right\|_F - q(\delta_1 + \delta_2 + \delta_3) \|X\|_F,
\end{aligned}
\tag{49}
$$

where the second step is by equation (42) and the spectral norm bound on $N$, and the last step is again by the triangle inequality. Then, we have that

$$
\left\| UU^\top X \right\|_F^2 \leq \left\| UU^\top \right\|_2^2 \cdot \|X\|_F^2 \leq (\sigma_{\max}(U))^4 \|X\|_F^2.
$$

On the other hand, $\left\| XX^\top X \right\|_F^2 = \sum_{i=1}^d \sigma_i^6$, where $\sigma_i$ is the $i$-th singular value of $X$, for $i = 1, 2, \ldots, r$. Notice that

$$
\sum_{i=1}^r \sigma_i^2 \leq \left( \sum_{i=1}^r \sigma_i^6 \right)^{\frac{1}{3}} r^{\frac{2}{3}} \Rightarrow \left( \sum_{i=1}^r \sigma_i^2 \right)^{\frac{3}{2}} \cdot r^{-1} \leq \left( \sum_{i=1}^r \sigma_i^6 \right)^{\frac{1}{2}}
$$

by Hölder's inequality. Thus, plugging in this fact into the above equation (49), we obtain that

$$
\|X\|_F^2 = \sum_{i=1}^r \sigma_i^2 \leq r \cdot \sigma_{\max}(U))^2 + (\delta_1 + \delta_2 + \delta_3)r \leq r \cdot (\sigma_{\max}(U))^2 + \delta,
$$

which implies that $\|X\|_F \leq r^{1/2}\sigma_{\max}(U) + \delta^{1/2}$. This concludes the proof of equation (43), which implies equation (44) based on equation (48).

Second, we look at the second-order optimality condition and show that this condition implies that the smallest singular value of $X$ is lower bounded from below.

**Lemma B.9.** *In the setting of Theorem 3.4, suppose $X$ satisfies the second-order optimality condition* (38) *and equation* (39). *Suppose $\alpha \geq 4\kappa r\sqrt{\mu/d}$ and $\lambda = \mu(r+1)q/\alpha^2$.*

*When $q \geq \frac{c\kappa^6\mu^2 r^3 \log(d)}{d}$ for a fixed constant $c > 0$, and $d$ is large enough, with probability at least $1 - O(d^{-1})$ over the randomness of $\Omega$, we have that*

$$(\sigma_{\min}(X))^2 \geq \frac{1}{8}(\sigma_{\min}(U))^2. \tag{50}$$

*Proof.* Let $A = \{i : \|X_i\| \leq \alpha\}$ be the index set of row vectors of $X$ whose $\ell_2$ norm is at most $\alpha$. Let $U_A$ be the matrix that has the same $i$-th row as the $i$-th row of $U$ for every $i \in A$, and 0 elsewhere. Let $v \in \mathbb{R}^r$ be a unit vector such that $\|Xv\| = \sigma_{\min}(X)$.

We show that

$$\sigma_{\min}(U_A) \geq (1 - (32\kappa)^{-1})\sigma_{\min}(U).$$

Let $B = \{1, 2, \ldots, d\} \setminus A$. Since for any $i \in B$, $\|X_i\| \geq \alpha$, we have that $|B|\alpha^2 \leq \|X\|_F^2 \leq (\sigma_{\max}(U))^2 r + \delta$ by equation (43). It also follows that $|B| \leq (\sigma_{\max}(U)^2 r + \delta)/\alpha^2$.

Next, we have that

$$\begin{aligned}
\sigma_{\min}(U_A) &= \sigma_{\min}(U - U_B) \\
&\geq \sigma_{\min}(U) - \sigma_{\max}(U_B) \\
&\geq \sigma_{\min}(U) - \|U_B\|_F \\
&\geq \sigma_{\min}(U) - \sqrt{|B|r\mu/d} \\
&\geq \sigma_{\min}(U) - \sqrt{\frac{(\sigma_{\max}(U)^2 r + \delta)r\mu}{d\alpha^2}} \\
&\geq (1 - \frac{1}{4})\sigma_{\min}(U),
\end{aligned} \tag{51}$$

where the last line is because $\alpha \geq 4\kappa r\sqrt{\mu/d}$. It also follows that $U_A$ has a column rank equal to $r$.

Notice that there must exist a unit vector $u_A$ in the column span of $U_A$ such that $\|X^\top u_A\| \leq \sigma_{\min}(X)$. To see this, we can simply decompose the column span of $U_A$ into one in the column span of $X$ and another into the orthogonal subspace. Since $u_A$ is a unit vector, we can also write $u_A$ as $u_A = U_A\beta$, for some $\beta \in \mathbb{R}^r$ where $\|\beta\| \leq \frac{1}{\sigma_{\min}(U_A)} \leq \frac{4}{3\sigma_{\min}(U)}$. Further,

$$\|u_A\|_\infty \leq \left(\max_{i=1}^d \|U_A e_i\|\right) \cdot \|\beta\| \leq \frac{4\sqrt{\mu r/d}}{3\sigma_{\min}(U)}.$$

Next, we plug in $V = u_A v^\top$ in the second-order optimality condition of equation (38). Note that since $u_A$ is in the column span of $U_A$, it is supported on the subset of columns spanned by $A$, and therefore $[\nabla^2 R(X)]V = 0$.[2] Therefore, the term about the Hessian of the regularization term in equation (38) is equal to zero. Thus, by taking $V = u_A v^\top$ in equation (38), we get that

$$\langle P_\Omega(u_A(Xv)^\top + Xvu_A^\top), u_A(Xv)^\top + Xvu_A^\top \rangle \geq 2u_A^\top P_\Omega(UU^\top - XX^\top)u_A + 2u_A^\top N u_A. \tag{52}$$

Note that

$$\|Xv\|_\infty \leq (\max_{i=1}^d \|X_i\|)\, \|v\| \leq 2\sqrt{\mu(r+1)q/\lambda} = 2\alpha,$$

---

[2]For details about the calculation regarding $\nabla^2 R(\cdot)$, see Lemma 18, Ge et al. (2017).

based on Lemma B.7 and the fact that $v$ is a unit vector. As a result,

$$\left\|Xvu_A^\top\right\|_\infty \leq \frac{8\alpha\sqrt{\mu r/d}}{3\sigma_{\min}(U)} := \frac{\delta_1}{d},$$

where $\delta_1 = \frac{8\kappa^2 r^{3/2}\mu}{3\sigma_{\min}(U)}$. By applying Proposition B.4 to the left-hand side of equation (52), we have that

$$\left|\langle P_\Omega(u_A(Xv)^\top + Xvu_A^\top), u_A(Xv)^\top + Xvu_A^\top\rangle - q\left\|u_A(Xv)^\top + Xvu_A^\top\right\|_F\right| \leq q\delta_1\sqrt{\frac{16\log(2d)}{dq}}.$$

Next, by applying Lemma 4.2 to the first term in the right-hand side of equation (52), we get

$$\left|u_A^\top P_\Omega(UU^\top - XX^\top)u_A - qu_A^\top(UU^\top - XX^\top)u_A\right| \leq q\delta_2\sqrt{\frac{16\log(2d)}{dq}},$$

where

$$\delta_2 = d\left\|UU^\top - XX^\top\right\|_\infty \leq \mu r + 4d\alpha^2 \leq 65\kappa\mu r.$$

Finally, we apply the spectral norm bound on $N$ from equation (B.3) to $u_A^\top N u_A$. Put together, we get

$$q\left\|Xvu_A^\top + u_A(Xv)^\top\right\|_F^2 \geq 2q\langle UU^\top - XX^\top, u_Au_A^\top\rangle - \delta q,$$

where

$$\delta = (\delta_1 + \delta_2)\sqrt{\frac{16\log(2d)}{dq}} + \sqrt{\frac{32\mu^2 r^2\log(2d)}{dq}} + \tilde{O}(n^{-1/2}d^{-1}).$$

By simplifying the above calculation and using the fact that $u_A$ is a unit vector, we get that

$$4\left\|Xv\right\|^2 + 2\left\|X^\top u_A\right\|^2 \geq 2\left\|U^\top u_A\right\|^2 - \delta. \tag{53}$$

Recall that $u_A$ is a unit vector and $u_A$ is in the column span of $U_A$. Therefore,

$$\|U^\top u_A\| = \|U_A^\top u_A\| \geq (\sigma_{\min}(U_A))^2,$$

and the right hand side of equation (53) is greater than $2(\sigma_{\min}(U_A))^2 - \delta$.

Moreover, recall that $\|Xv\| = \sigma_{\min}(X)$ and $\|X^\top u_A\| \leq \sigma_{\min}(X)$. Therefore, the left hand side of equation (53) is at most $6(\sigma_{\min}(X))^2$, in which we apply Cauchy-Schwarz to the cross term. Put together, from equation (53) we conclude that

$$(\sigma_{\min}(X))^2 \geq \frac{1}{3}(\sigma_{\min}(U_A))^2 - \frac{\delta}{6} \geq \frac{3}{16}(\sigma_{\min}(U))^2 - \frac{\delta}{6},$$

by equation (51).

Thus, when

$$dq \geq c\kappa^6 r^3\mu^3\log(d),$$

for $c = 16 \times 65^2 \times 9$, we have that

$$\frac{\delta}{6} \leq \frac{\sigma_{\min}(U))^2}{16},$$

which concludes the proof of equation (50).

## B.5 Characterization of The Population Loss

In the next result, we show that we could remove the effect of the regularizer term in the residual error. We use the fact that the regularizer follows the same direction as $X$, as stated in Proposition B.6.

**Lemma B.10.** *In the setting of Theorem 3.4, suppose $X \in \mathbb{R}^{d \times r}$ satisfies that*

$$(\sigma_{\min}(X))^2 \geq \frac{1}{8}(\sigma_{\min}(U))^2,$$

*and $\alpha \geq 4\kappa^2 r \sqrt{\mu/d}$,*

$$\left\| UU^\top X - XX^\top X \right\|_F^2 \leq \left\| UU^\top X - XX^\top X - \gamma \nabla R(X) \right\|_F^2, \quad \text{for any } \gamma \geq 0. \tag{54}$$

*Proof.* Let $S = \{ i \in [d] : |X_i| \geq \alpha \}$ be the index set of row vectors of $X$ whose $\ell_2$ norm is greater than $\alpha$. By the definition of the regularizer $R(X)$, for any $i$ not in $S$, we have that $(\nabla R(X))_i = 0$. Thus, for any $i \notin S$, we have that

$$\left\| U_i U^\top X - X_i X^\top X \right\| = \left\| U_i U^\top X - X_i X^\top X + (\nabla R(X))_i \right\|.$$

For any $i$ that is in $S$, we have that

$$\left\| U_i U^\top X - X_i X^\top X \right\|^2 = \left\| U_i U^\top X - X_i X^\top X - (\gamma \nabla R(X))_i \right\|^2,$$

where $U_i, X_i$ are the $i$-th row vectors of $U, X \in \mathbb{R}^{d \times r}$, for every $i = 1, 2, \ldots, d$. As for each $i \in S$, we will show that

$$\langle (\nabla R(X))_i, X_i X^\top X - U_i U^\top X \rangle \geq 0. \tag{55}$$

By Proposition B.6, we have

$$(\nabla R(X))_i = \left( 4\lambda \sum_{i=1}^{d} \frac{(|X_i|^3 - \alpha)_+^3}{|X_i|} \right) X_i := a_i X_i,$$

where $a_i \geq 0$ denotes the multiplier in front of $X_i$. Then, we have that

$$
\begin{aligned}
\langle (\nabla R(X))_i, X_i X^\top X \rangle &= a_i \left\| X X_i^\top \right\|^2 \\
&\geq a_i \left\| X_i \right\|^2 \sigma_{\min}(X^\top X) \\
&\geq \frac{a_i}{8} \left\| X_i \right\|^2 (\sigma_{\min}(U))^2,
\end{aligned}
$$

by the condition on the smallest singular value of $U$. On the other hand,

$$
\begin{aligned}
\langle (\nabla R(X))_i, U_i U^\top X \rangle &= a_i \langle X_i, U_i U^\top X \rangle \\
&\leq a_i \left\| X_i \right\| \cdot \left\| U_i \right\| \cdot \sigma_{\max}(U) \cdot \sigma_{\max}(X) \\
&\leq a_i \left\| X_i \right\| \left\| U_i \right\| (\sigma_{\max}(U))^2 2\sqrt{r} \qquad \text{(by equation (43), plus } \sigma_{\max}(X) \leq \|X\|_F) \\
&\leq a_i \left\| X_i \right\| \left\| U_i \right\| (\sigma_{\min}(U))^2 2\kappa^2 \sqrt{r} \qquad \text{(since } \sigma_{\max}(U) = \kappa \cdot \sigma_{\min}(U)) \\
&\leq \frac{a_i}{16} \left\| X_i \right\|^2 (\sigma_{\min}(U))^2,
\end{aligned}
$$

where the last step is because

$$\|X_i\| \geq \alpha \geq 4\kappa^2 r \sqrt{\frac{\mu}{d}} \geq 4\kappa^2 \sqrt{r} \left\| U_i \right\|,$$

since $\|U_i\| \leq \sqrt{\mu r/d}$ by Assumption 3.3. Putting both sides together, we therefore conclude that equation (55) must hold. The proof is completed. □

Finally, we show that the above result also implies that $UU^\top$ is close to $XX^\top$.

**Lemma B.11.** *In the setting of Theorem 3.4, suppose $X \in \mathbb{R}^{d \times r}$ satisfies equations (44) and (54). Then, with probability at least $1 - O(d^{-1})$, the following must be true:*

$$\left\| XX^\top - UU^\top \right\|_F^2 \leq \frac{c\mu^2 r^5 \kappa^6 \log(2d)}{dq}, \tag{56}$$

*for some fixed constant $c > 0$.*

*Proof.* We separate $U$ into one part that is in the column span of $X$ and another part that is orthogonal to the column span of $X$. Let $U = Z + V$ where $Z$ is the projection of $U$ to the column span of $X$, and $V$ is the projection of $U$ to the orthogonal column subspace of $X$. In particular, we have $V^\top X = 0$. Thus,

$$UU^\top X = (Z + V)(Z + V)^\top X = ZZ^\top X + VZ^\top X$$
$$\Rightarrow \left\| UU^\top X - XX^\top X \right\|^2 = \left\| ZZ^\top X - XX^\top X \right\|_F^2 + \left\| VZ^\top X \right\|_F^2 \leq \delta^2, \tag{57}$$

for

$$\delta = 42\sigma_{\max}(U)\sqrt{\frac{\mu^2 r^4 \log(2d)}{dq}},$$

by applying equation (44) and also equation (54).

Notice that $Z$ has the same column span as $X$. As a result,

$$\left\| ZZ^\top X - XX^\top X \right\|_F^2 \geq \sigma_{\min}^2(X) \left\| ZZ^\top - XX^\top \right\|_F^2,$$

which can be verified by inserting the SVD of $X$ into the above inequality. Combined with equation (57), we thus have that

$$\left\| ZZ^\top - XX^\top \right\|_F^2 \leq \frac{\delta^2}{(\sigma_{\min}(X))^2} \leq \frac{8\delta^2}{(\sigma_{\min}(U))^2},$$

since $(\sigma_{\min}(X))^2 \geq (\sigma_{\min}(U))^2/8$.

Next, let $a \in \mathbb{R}^d, b \in \mathbb{R}^r$ be two unit vectors such that

$$a^\top ZZ^\top Xb = \sigma_{\min}(ZZ^\top X).$$

From equation (57) we can infer that

$$\left| \sigma_{\min}(ZZ^\top X) - a^\top XX^\top Xb \right| \leq \delta. \tag{58}$$

Now we consider two cases. Suppose $a^\top XX^\top Xb$ is positive. Then the left-hand side of equation (58) must be at least

$$a^\top XX^\top Xb - \sigma_{\min}(ZZ^\top X) = \sigma_{\min}(XX^\top X) - \sigma_{\min}(ZZ^\top X).$$

The other case is that if $a^\top XX^\top Xb$ is negative, then the left hand side of equation (58) must be at least

$$\left| a^\top XX^\top Xb \right| - \sigma_{\min}(ZZ^\top X) \geq \sigma_{\min}(XX^\top X) - \sigma_{\min}(ZZ^\top X).$$

We know that

$$\sigma_{\min}(XX^\top X) \geq (\sigma_{\min}(U))^3/(8\sqrt{8}).$$

Thus, if $\delta \leq (\sigma_{\min}(U))^3/(16\sqrt{8})$, then

$$\sigma_{\min}(ZZ^\top X) \geq (\sigma_{\min}(U))^3/(16\sqrt{8}).$$

On the other hand,

$$\sigma_{\min}(ZZ^\top X) \leq \sigma_{\max}(Z) \cdot \sigma_{\min}(Z^\top X)$$
$$\leq \|U\|_2 \cdot \sigma_{\min}(Z^\top X),$$

which implies

$$\sigma_{\min}(Z^\top X) \geq \frac{(\sigma_{\min}(U))^3}{8\sqrt{8}\, \|U\|_2}.$$

Also note that $\left\|VZ^\top X\right\|_F^2 \leq \delta^2$, which implies that

$$\|V\|_F^2 \leq \frac{\delta^2}{(\sigma_{\min}(Z^\top X))^2} \leq \frac{512\delta^2 \|U\|_2^2}{(\sigma_{\min}(U))^6} = \frac{512\delta^2\kappa^2}{(\sigma_{\min}(U))^4}.$$

Finally,

$$\left\|ZV^\top\right\|_F \leq \|Z\|_2 \cdot \|V\|_F \leq \|U\|_2 \cdot \|V\|_F,$$

since $Z$ is a projection of $U$ to the column space of $X$.

In summary, we have that

$$
\begin{aligned}
&\left\|UU^\top - XX^\top\right\|_F^2 \\
&= \left\|ZZ^\top - XX^\top\right\|_F^2 + 2\left\|ZV^\top\right\|_F^2 + \left\|VV^\top\right\|_F^2 \\
&\leq \frac{16\delta^2}{(\sigma_{\min}(U))^2} + \frac{1024\delta^2\kappa^4}{(\sigma_{\min}(U))^2} + \frac{512^2\delta^4\kappa^4 r}{(\sigma_{\min}(U))^8} \\
&\leq \frac{(512\kappa^4 r + 1024\kappa^4 + 16)\delta^2}{(\sigma_{\min}(U))^2},
\end{aligned}
$$

since $\delta \leq (\sigma_{\min}(U))^3/(8\sqrt{8})$, which concludes the proof of equation (56), for $c = 42^2 \times 512$.

Now we are ready to complete the proof of Theorem 3.4.

*Proof of Theorem 3.4.* Suppose $X$ satisfies the first-order and second-order optimality conditions. Then by Lemma B.7 and Lemma B.8, we have that $X$ satisfies the incoherence condition in equation (39). As a result, $(\sigma_{\min}(X))^2 \geq (\sigma_{\min}(U))^2/8$ by Lemma B.9. In particular, we require $\alpha$ to be at least $4\kappa^2 r\sqrt{\mu/d}$.

In addition, based on Lemma B.8, we require $\alpha \leq \sqrt{\mu(r+1)q/\lambda}$, which requires

$$\lambda \geq \mu(r+1)q/\alpha^2 \geq dq/8.$$

Recall that

$$q = 1 - (1 - p^2)^n = 1 - (1 - C^2/d^2)^n \approx C^2 n/d^2.$$

When

$$n \geq \frac{cdr^5\kappa^6\mu^2\log(2d)}{C^2\epsilon^2},$$

the right hand side of equation (56) is at most $\epsilon^2$. This concludes the proof of equation (10).

**Remark B.12.** *Our proof has been inspired by the important early work of Ge et al. (2016). We build on and expand their work in three aspects.*

- *First, in the one-sided completion setting, the sampling patterns are non-random. In particular, the diagonal entries are fully observed, while the off-diagonal entries are sparsely observed. Thus, there is a mix of dense and sparse samplings in $\Omega$. To address this problem, we use a weighted projection as defined in Section 2.*

- *Second, the bias of the Hájek estimator is complex due to the nonlinear form of the estimator, and requires a delicate analysis of the noise matrix $N$. This is studied in detail in Corollary B.2 and Corollary B.3.*

- *Third, the observation patterns in the off-diagonal entries of $\Omega$ are not completely independent, requiring a new, row-by-row concentration inequality that leverages a conditional independence property, i.e., Lemma 4.2.*

*More broadly, non-uniform sampling patterns are very common in real-world datasets and have received lots of recent interest in causal inference (Athey et al., 2021; Xiong & Pelger, 2023). We hope this work inspires further studies of modeling non-uniform sampling on sparse panel data from an optimization perspective.*

*Recent work such as Dong et al. (2025) examine the optimization landscape of neural networks through evaluating the Hessian structure, such as the Hessian spectrum. In light of the proof techniques developed in this work, it might also be interesting to revisit these insights in the setting of low-rank matrix factorization, which can be related to two-layer quadratic neural networks (Li et al., 2018).*

## C Omitted Experiments

In this section, we describe the dataset statistics and pre-processing procedures. The MovieLens-32M dataset comprises 32 million user ratings on approximately $87,000$ movies, collected from the MovieLens recommendation platform, with ratings ranging from 1 to 5. The sparsity ratio is roughly $3 \times 10^{-4}$. We also consider two related MovieLens datasets with 20 million and 25 million entries to ensure the robustness of our findings. We split 80% into training and hold out the rest for testing.

- The Amazon Reviews dataset includes 571 million user ratings on 48 million products, collected from Amazon, with ratings ranging from 1 to 5. The sparsity ratio is roughly $2 \times 10^{-7}$. For the Amazon reviews dataset, we choose the Automotive category with $50,000$ items rated by $100,000$ users as $M$. We use 80-20 split.

- The Genomes dataset contains phased genotype data for $2,054$ individuals gathered from diverse populations. The dataset is represented as a dense matrix, with rows representing genomic sites and columns corresponding to individuals. Each entry represents a biallelic genotype, encoded as 0 or 1. We map them to 1 and 2, and use 0 to indicate missing entries. This dataset is fully observed. Thus, we use 1% for training and hold out the rest for testing.

- For MovieLens and Amazon Reviews datasets, we use $\widehat{T}$ on $M^\top M$ to obtain the ground truth second-moment matrix. Then, the estimation error is evaluated against this matrix. In our experiments, we set the sampling probability $p = 0.8$. Given that the original datasets have an average sparsity of approximately 0.3%, even with $p = 0.8$, the number of observed entries remains very sparse. When $T$ is not fully observed, we focus on measuring the mean squared error within the observed entries instead and treat the missing entries as zero.

### C.1 Implementation

We provide a detailed description of all the baseline methods we have implemented in our experiments.

- Alternating-GD minimizes the squared reconstruction error on the observed entries of $M$. We minimize the following objective:

$$\min_{X \in \mathbb{R}^{n \times r}, Y \in \mathbb{R}^{d \times r}} \|P_\Omega(XY^\top - M)\|_F^2,$$

where $P_\Omega$ denotes the projection onto the observed entries. We initialize $X$ and $Y$ from an isotropic Gaussian distribution. We perform the optimization using Adam with a learning rate of 0.1 over 300 training epochs.

- SoftImpute-ALS alternates between matrix completion using singular value decomposition (SVD) and imputing missing values (Hastie et al., 2015). At each iteration, missing entries are filled with the current estimates, followed by SVD and soft thresholding of the singular values. The resulting

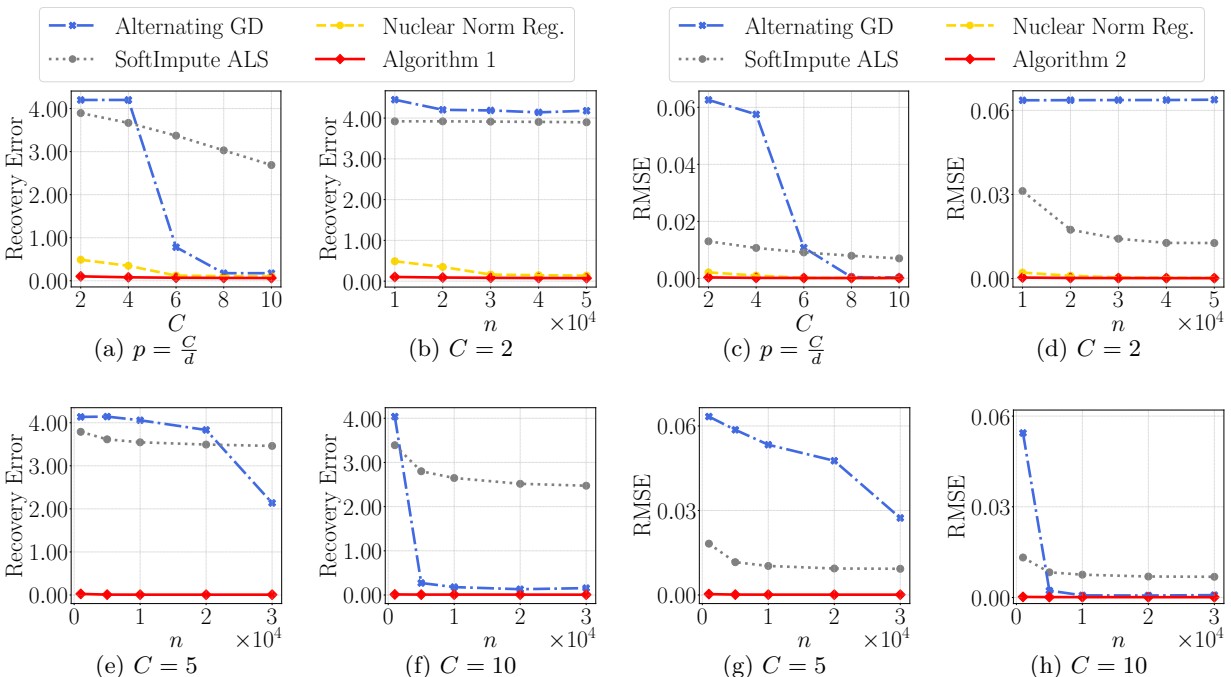

Figure 7: One-sided recovery error of $T$ (shown in the left two columns) and the root mean squared recovery error of the original matrix $M$ (shown in the right two columns) using our algorithms on synthetic matrix data. Figures 7a, 7b: One-sided recovery error under uniform sampling by varying $p$, and sampling two entries per row ($C = 2$) while varying the number of rows $n$. Figures 7c, 7d: Recovery error on $M$, also under uniform sampling, and sampling two entries per row. Overall, we find that our approach consistently provides more accurate estimates compared with the three baseline methods, all of which are widely used in the matrix completion literature. Figures 7e, 7f, 7g, 7h: Vary $C$ and repeat the same experiment.

truncated low-rank approximation is used to update the matrix. We repeat this process until convergence, as measured by the Frobenius norm difference between successive reconstructions, or until a maximum of 500 iterations. We set the target rank to the same as $r$ and the convergence threshold to $10^{-5}$.

- Nuclear norm regularization is another commonly used approach for matrix completion. In one-sided estimation, we compute empirical averages based on the observed data (Cao et al., 2023). For off-diagonal entries $(i, j)$, we identify all samples in which the pair $(i, j)$ appears and average the product of the corresponding entries from the two relevant columns. For diagonal entries $(i, i)$, we consider all samples where index $i$ appears in either position of a pair, square the corresponding data matrix entry for each such sample, and average these squared values. We then estimate the full matrix using symmetric alternating gradient descent by solving the following optimization problem:

$$\min_{X \in \mathbb{R}^{d \times r}} \left\| P_\Omega(XX^\top - T) \right\|_F^2 + \lambda \left\| X \right\|_\star,$$

where $\|X\|_\star$ denotes the nuclear norm of $X$ and $\lambda$ is set as 0.01. We perform the optimization using Adam with a learning rate of 0.1 for $1,000$ steps.

*Hyperparameter configurations.* We report the hyperparameter configurations used in our experiments. The learning rate $\eta$ is varied from 10 to $10^5$, the regularization coefficient $\lambda$ from $10^{-2}$ to $10^{-5}$, and the regularization parameter $\alpha$ from $10^{-4}$ to 1. Based on cross-validation, we select $\eta = 10^4$, $\lambda = 10^{-4}$, and $\alpha = 10^{-3}$, which yield the lowest error. For the Amazon Review dataset, we set $\lambda = 0.01$ and $\alpha = 0.1$.

Table 4: We report the recovery error for both one-sided matrix completion (i.e., recovering $T$), and recovering $M$, along with the corresponding running time (measured in seconds on an Ubuntu server) on three MovieLens datasets. We run each experiment with five random seeds and report the mean and standard deviation.

| Dataset | MovieLens-20M | MovieLens-25M | MovieLens-20M | MovieLens-25M |
|---|---|---|---|---|
| Estimating $T$ | One-sided recovery error | | Running time (Seconds) | |
| Alternating-GD | $0.015_{\pm 0.009}$ | $0.006_{\pm 0.002}$ | $1.9_{\pm 0.1} \times 10^2$ | $4.4_{\pm 0.1} \times 10^2$ |
| SoftImpute-ALS | $0.008_{\pm 0.005}$ | $0.006_{\pm 0.002}$ | $3.4_{\pm 0.1} \times 10^3$ | $9.4_{\pm 0.7} \times 10^3$ |
| Algorithm 1 | $\mathbf{0.002_{\pm 0.001}}$ | $\mathbf{0.002_{\pm 0.10}}$ | $\mathbf{4.8_{\pm 0.2} \times 10^1}$ | $\mathbf{1.1_{\pm 0.6} \times 10^2}$ |
| Estimating $M$ | Root mean squared error | | Running time (Seconds) | |
| Alternating-GD | $1.11_{\pm 0.00}$ | $1.27_{\pm 0.02}$ | $1.9_{\pm 0.1} \times 10^2$ | $4.4_{\pm 0.1} \times 10^2$ |
| SoftImpute-ALS | $1.09_{\pm 0.00}$ | $1.25_{\pm 0.00}$ | $3.4_{\pm 0.1} \times 10^3$ | $9.4_{\pm 0.7} \times 10^3$ |
| Algorithm 2 | $\mathbf{0.99_{\pm 0.00}}$ | $\mathbf{1.04_{\pm 0.00}}$ | $\mathbf{1.0_{\pm 0.1} \times 10^2}$ | $\mathbf{2.6_{\pm 0.1} \times 10^2}$ |

## C.2 Omitted Comparison Results

First, we report the runtime comparison between HÁJEK-GD and nuclear norm regularization. We run gradient descent with $1,000$ iterations until it has converged. We also fix the number of observed entries as $200d$ while varying $d$ from $10^3$ to $10^4$. We find that our approach requires $0.95$ seconds for $d = 10^3$ and $1.61$ seconds for $d = 10^4$. By contrast, solving a convex program with nuclear norm penalty takes $5.68$ seconds for $d = 10^3$ and $100.34$ seconds for $d = 10^4$, measured on the Ubuntu server.

Second, we present the recovery error for imputing $M$, illustrated in Figure 7. Our approach consistently reduces the recovery error compared to all three baseline methods. For instance, when $C = 2$ (sampling two entries per row), our approach incurs an error of $0.0003$, compared to $0.002$ for nuclear norm regularization. We also observe similar results on synthetic data when sampling five entries per row ($C = 5$) or sampling ten entries per row ($C = 10$) in Figures 7e and 7f.

Similar results are observed when our approach is applied to recover the missing entries of $M$. For recovering $M$, our approach reduces estimation error by $94\%$ relative to alternating-GD and by $98\%$ compared to softImpute-ALS.

Finally, we report the results on MovieLens-20M and MovieLens-25M in Table 4. For one-sided matrix completion, our approach reduces the Frobenius norm error between the estimate and the true $T$ by up to $87\%$ compared to alternating-GD and up to $75\%$ compared to softImpute-ALS, while also achieving lower running times.

For recovering the missing entries of $M$, our approach reduces the RMSE by up to $10\%$ compared to alternating-GD and up to $6\%$ compared to softImpute-ALS.

## C.3 Ablation Analysis

*Setting the rank of the target matrix.* In Section 5, we use a target rank of $10$ for all the reported experiments. To further assess the robustness of our approach, we conduct additional experiments on three real-world datasets using Algorithm 1, with the target rank in the range of $1$ to $30$. The results are shown in Table 5. We find that larger ranks may lead to overfitting in sparse settings; for example, on the Amazon dataset, increasing $r$ beyond $10$ results in worse performance when recovering $M$ from the estimated $T$.

We also run similar analyses on synthetic data and find that the algorithm is not particularly sensitive to different choices in this context. In particular, we conduct different simulations by varying the rank $r$ of $M$ between $1$ and $50$. We vary the target rank used in Algorithm 1 between $1$ and $50$. We find that the estimation errors are comparable between different choices of target ranks.

Table 5: We vary the rank of the variable matrix in HÁJEK-GD, corresponding to the results we reported in Table 3. The results are averaged over five independent runs.

| Dataset | MovieLens-20M | Amazon Reviews | Genomes |
|---|---|---|---|
| Estimating $T$ (Using Algorithm 1) | One-sided recovery error on $T$ | | |
| $r = 1$ | $7.0_{\pm 0.1} \times 10^{-3}$ | $4.5_{\pm 0.1} \times 10^{-3}$ | $5.8_{\pm 0.1} \times 10^{-5}$ |
| $r = 10$ | $4.7_{\pm 0.1} \times 10^{-3}$ | $\mathbf{1.3}_{\pm 0.1} \times 10^{-3}$ | $\mathbf{5.8}_{\pm 0.1} \times 10^{-5}$ |
| $r = 20$ | $1.9_{\pm 0.1} \times 10^{-3}$ | $1.3_{\pm 0.1} \times 10^{-3}$ | $5.8_{\pm 0.1} \times 10^{-5}$ |
| $r = 30$ | $\mathbf{1.8}_{\pm 0.1} \times 10^{-3}$ | $1.3_{\pm 0.1} \times 10^{-3}$ | $5.8_{\pm 0.1} \times 10^{-5}$ |
| Estimating $M$ (Using Algorithm 2) | Root mean squared recovery error on $M$ | | |
| $r = 1$ | $3.7_{\pm 0.1}$ | $4.6_{\pm 0.1}$ | $\mathbf{0.1}_{\pm 0.1}$ |
| $r = 10$ | $1.1_{\pm 0.1}$ | $\mathbf{1.9}_{\pm 0.1}$ | $0.2_{\pm 0.1}$ |
| $r = 20$ | $\mathbf{1.0}_{\pm 0.1}$ | $2.3_{\pm 0.1}$ | $0.2_{\pm 0.1}$ |
| $r = 30$ | $1.0_{\pm 0.1}$ | $2.7_{\pm 0.1}$ | $0.2_{\pm 0.1}$ |

*Sensitivity of Algorithm 1.* We evaluate the sensitivity of adding noise to $\hat{M}$ on the final output of Algorithm 1. We use synthetic datasets with varied numbers of rows between $5d$ and $20d$, and a fixed dimension of $1,000$. We also test different ranks of $M$, ranging from 5 to 20. We add Gaussian noise with a standard deviation that ranges from 0 to 0.01 to the input. Interestingly, we find that estimation error increases linearly with $\sigma$. Moreover, when varying $n$, the sensitivity level of HÁJEK-GD (recall its definition in equation (17)) is generally low. When $n = 5d$, the slope of the line is 19. When $n = 10d$, the slope decreases to 14. When $n = 20d$, the slope further decreases to 13. Similar results hold after varying the rank between $20, 10, 5$, and the sensitivity level ranges between $16, 13, 11$. See illustration in Figure 8.

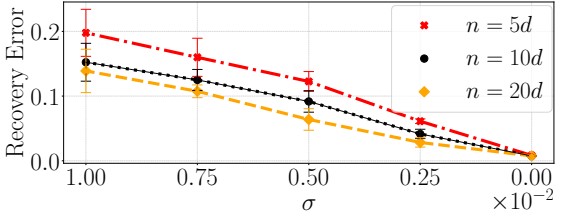

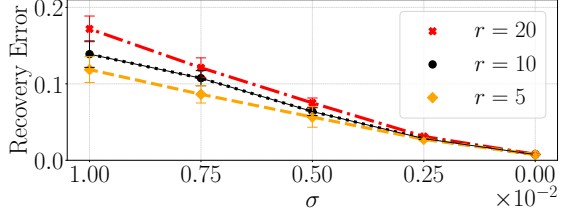

(a) Varying noise level $\sigma$ for different number of rows $n$     (b) Varying noise level $\sigma$ for different rank $r$ of $M$

Figure 8: Illustrating the estimation error after adding Gaussian noise with mean zero and variance $\sigma^2$ to $\hat{M}$ on the non-zero entries. In Figure 8a, we vary the number of rows $n$ with a fixed rank $r = 10$. In Figure 8b, we vary rank $r$ (of $M$) with a fixed $m = 20d$. Across all six cases, the sensitivity level, measured by the slope of the line, is at most 19 and drops to 11 as the sample size increases or the rank decreases. We report the mean and standard deviation from five independent runs.

