# OpenReview forum: "One-Sided Matrix Completion from Ultra-Sparse Samples"
_TMLR — Accepted by TMLR_

### Review · Reviewer_PeU4 · 2025-08-12

**Summary Of Contributions:**

This paper studies the matrix completion problem when the number of samples being taken is extremely small, only $Cn$. In this setting, recover the matrix $M$ is not possible, so authors consider recovering the second moment $T=\frac{1}{n}M^\top M$. The major challenge here is that the binary mask matrix $\Omega$ for $T$ is not uniform, as it has much higher probabilities on the diagonal than the off-diagonals. To address this issue, the authors introduce the Hajek estimator which obtains lower variance than the more commonly-used HT estimator. And then, they show that use a properly regularized objective, one could recover $T$ up to $\epsilon^2$ error in squared Frobenius norm via first compute the Hayek estimator, then run gradient descent and use the entries outside of $\Omega$ obtained by gradient descent. Extensive experiments are performed to show that Hayek estimator indeed has lower variance than HT estimator on synthetic dataset. Further, on real-world datasets, the proposed Hajek-GD algorithm outperforms methods such as alternating GD and softImpute ALS in both recovery error and runtime.

**Audience:**

Yes

**Broader Impact Concerns:**

No such concerns.

**Claims And Evidence:**

Yes

**Requested Changes:**

Major: see weaknesses, particularly the major weakness.

Minor typos:

* Hja&#769;ke has its punctuation misplaced in multiple places, sometimes it's missed, sometimes it's typed as Hj&#769;ake. I suggest authors to carefully proofread the manuscript and make sure the use of punctuation is consistent.

* In Theorem 3.4, it might be better to first remind readers the definition of $\alpha$ and $\lambda$ for regularization before setting their values.

**Strengths And Weaknesses:**

Strengths:

I think this is a strong paper overall. It studies an important problem: matrix completion when one could only observe a constant number of entries per row. To tackle this challenge, authors propose to use Hajek estimator, which is novel in the matrix completion literature. They conduct rigorous theoretical studies on Hajek estimator for matrix completion, prove its conditional unbiasedness and variance reduction from HT estimator. This is not particularly easy as Hajek estimator involves dividing random variables. To do so they analyze the variance via first-order approximation. I think the introduction of Hajek estimator to the matrix completion community is a nice contribution. Experiments are comprehensive, they cover various aspects (variance, relationship between different parameters and $d$, recovery error and runtime).

Weaknesses:

There are two weaknesses, one major, one minor.

1 (Major). The "common sense" assumption when analyzing the Hajek-GD algorithm. This model assumes the rows of $M$ are drawn uniformly from $r$ vectors $u_1,\ldots,u_r$. This seems to be a very restrictive assumption --- in particular, this means the other rank-$r$ factor is essentially rearranging and duplicating columns of the factor matrix $U$. This deserves many more discussions than a footnote. Further, the references justifying this assumptions are all related to regressions, I assume this means that this assumption directly impacts the proof of the gradient descent step. Authors should thoroughly discuss and justify this setting, how strongly the proofs are "dependent" on the assumption, etc.

2 (Minor). One of the major advantages of gradient descent is its efficiency, authors might consider conduct a preliminary asymptotic runtime analysis of Hajek-GD.

---

> ### Author Response · Authors · 2025-08-21
> **Response to Reviewer PeU4**
>
> Thanks for the detailed comments. Please find our response to each comment below.
>
> **Comment 1:** “The "common means" assumption when analyzing the Hajek-GD algorithm. Authors should thoroughly discuss and justify this setting, how strongly the proofs are "dependent" on the assumption, etc.”
>
> **Response 1:** Thanks for your insightful question. The common means model is often used in the statistical learning literature to study heterogeneous datasets, such as multitask learning or distributed learning. An early work in this literature, Kolar et al. (JMLR, 2011), studies the same model as ours for the problem of union support recovery in multitask learning. A more recent work, Dobriban and Sheng (JMLR, 2021), studies a similar model in the case of distributed ridge regression. Both papers assume that a set of common mean factors exists across a large set of heterogeneous datasets. We have now added a detailed Remark 3.5 at the end of Section 3 to better discuss the prior work that motivates us to focus on this model.
>
> One natural extension, which can be handled using our current proof, is when each row $M_i$ is additionally perturbed by a noise vector with mean zero and variance $\sigma^2$. This can be achieved as we now just need to incorporate the noise variance into the error bounds when we analyze the bias of $\hat T$. In Remark 3.5, we have now added the steps required to modify the proof in order to handle this extension.
>
> Another plausible extension is when each row $M_i$ is a linear combination of the $r$ factors $\mu_1, \mu_2, \dots, \mu_r$. This setting appears to be much more difficult than the above common means model and would require the design of new estimators that are capable of first learning the underlying low-rank subspace from the linear combinations. One promising direction for tackling this extension is to examine spectral methods such as those used in the work of Chen et al. (Learning mixtures of low-rank models, IEEE Transactions on Information Theory, 2021). This would be an interesting (and likely significantly more challenging) question for future work.
>
> We hope these discussions address the reviewer’s question regarding the common-means model. It is also worth clarifying that this model is only posed when we prove the sample complexity result in Section 3.2. Notice that the unbiasedness of the Hajek estimator and its variance approximation both hold more generally, and are not restricted to this particular model.
>
> **Comment 2:** “One of the major advantages of gradient descent is its efficiency, authors might consider conduct a preliminary asymptotic runtime analysis of Hajek-GD.”
>
> **Response 2:** Thanks for your insightful comment. We have now included a runtime analysis of our algorithm in Section 5.3. In particular, in this experiment, we plot the runtime of Hajek-GD on both synthetic and real-world datasets. We find that the runtime generally scales linearly with the matrix dimensions, such as the number of columns $d$ and the number of observed entries in $\hat T$. Further analyzing the runtime precisely is an interesting question for future work.
>
> **Response to minor comments:** Thanks very much for your careful reading. We have now revised the paper to address the misspellings. We have also added more descriptions to clarify the notations before Theorem 3.4.
>
> Please let us know if you have additional comments or concerns about our paper during the discussion stage.

---

### Review · Reviewer_8mat · 2025-08-17

**Summary Of Contributions:**

This paper addresses the problem of one-sided matrix completion in the ultra-sparse regime, where only a small subset of entries from a tall matrix $M \in \mathbb{R}^{n \times d}$ (with $n \gg d$) are observed. Specifically, it focuses on recovering the row space of $M$ by estimating its second-moment matrix $T = \frac{1}{n} M^\top M$, despite extreme sparsity in the observations.

The proposed method, namely Hájek-GD, estimates the second-moment matrix $T$ in two steps. In the first step, for each pair of columns in $M$ that are co-observed in at least one row, it computes the average inner product between the corresponding column entries, normalized by the number of such co-observations. This yields a Hájek-type estimator that adjusts for the non-random and sparse observation pattern. However, due to extreme sparsity, many column pairs may never be jointly observed, leaving large portions of the second-moment matrix $T$ missing. To address this, the second step imputes the unobserved entries by fitting a low-rank factorization via gradient descent, with an incoherence regularization term to stabilize optimization and promote identifiability. The completed second-moment matrix, denoted by $\hat{T}$, is then used to recover the row span of $M$ via gradient descent.

The paper makes two main technical contributions. First, it demonstrates that the Hájek-type estimator is unbiased in finite samples and achieves substantially lower variance than the Horvitz-Thompson estimator, with greater variance reduction in sparser regimes. Second, it establishes a near-optimal recovery guarantee for the full second-moment matrix under a low-rank factor model with an incoherence condition.

Extensive experiments on real-world datasets, including MovieLens, Amazon Reviews, and 1000 Genomes, demonstrate that the proposed Hájek-GD method significantly outperforms standard matrix completion baselines in recovering both the second-moment matrix and the original matrix.

**Audience:**

Yes

**Claims And Evidence:**

Yes

**Requested Changes:**

- In the recovery of row span of $M$ from the completed matrix $\hat{T}$, the paper uses both diagonal and off-diagonal entries (as in Equation (8)). However, prior work on covariance and second-moment matrix estimation (e.g., Fan, Liao, and Mincheva, 2013) often treats the diagonal separately under a low-rank plus diagonal structure. It would be helpful to clarify under what conditions it is appropriate to include both diagonal and off-diagonal entries when estimating the low-rank component (i.e., the row span of $M$).
- It would be helpful to provide a precise and explicit definition of the "one-side matrix completion" problem in both the abstract and the introduction. In addition, it would be helpful to add some discussion about why this problem is important and challenging.
- Referring to $M$ as a tall matrix and stating that the goal is to estimate its row span would help concretize the problem setup for the reader.
- In Theorem 3.4, it would be helpful to discuss the assumption that the number of users $n$ must be much larger than the number of items $d$, and explain why this assumption is reasonable in real-world applications (e.g., in recommendation systems).
- The Hájek estimator is known to exhibit finite-sample bias in general. Since the paper shows it is unbiased in this setting, it would be helpful to highlight and explain this result more clearly in the introduction.
- The first extension, recovering individual entries of $M$, is closely aligned with the standard matrix completion problem. Therefore, it would be helpful to expand this extension a bit and discuss the practical scenarios where this extension is feasible (e.g., when the number of observed entries per row exceeds the rank of $M$).
- The second extension, concerning differential privacy, is currently a bit hard to follow and would benefit from revision for clarity.

Minor Issues and Typos
- In Figure 3, Step 1, the dimension of $\hat{T}$ is $d$ by $d$ as opposed to $n$ by $d$.
- The notation $S_i$ in Lemma 4.1 lacks a definition.
- “Bias of Hájek’s estimator” in the paragraph header on page 8 should be “Bias and Variance of Hájek’s estimator”.
- Hajek’s estimator -> Hajek estimator.
- Horvitz-Thompson’s estimator -> Horvitz-Thompson estimator.

**Strengths And Weaknesses:**

Strengths

- The paper tackles a challenging and practically important problem: matrix completion in the ultra-sparse regime, which arises frequently in large-scale recommendation systems where the number of users and items increases rapidly.
- The proposed Hájek-GD method is novel and well-motivated. Theoretical analysis of Hájek-type estimators in this context is nontrivial, and the paper provides rigorous and insightful results, including variance reduction and recovery guarantees.
- The empirical evaluation is thorough, covering a number of real-world datasets, and demonstrates substantial improvements over standard matrix completion baselines in terms of bias, variance, and recovery accuracy.

Weaknesses
- The paper is dense and presents a large number of technical results. The exposition could be improved to make the paper more accessible to a general audience.

---

> ### Author Response · Authors · 2025-08-21
> **Response to Reviewer 8mat**
>
> Thanks to the reviewer for detailed and constructive comments! We have thoroughly revised the paper based on the suggestions, and below we respond to each comment in detail.
>
> **Comment 1:** “In the recovery of row span, the paper uses both diagonal and off-diagonal entries. However, prior work on covariance and second-moment matrix estimation (e.g., Fan, Liao, and Mincheva, 2013) often treats the diagonal separately under a low-rank plus diagonal structure. It would be helpful to clarify under what conditions it is appropriate to include both diagonal and off-diagonal entries.”
>
> **Response to Comment 1:** Thanks for suggesting the related work of Fan, Liao, and Mincheva (2013). Notice that in their paper, they assume that the observed outcome $y$ follows a low-rank plus noise model. In particular, the noise introduces a bias in the covariance matrix, and the diagonal structure in their estimator is used to offset the bias incurred due to the variance of the noise in their setting.
>
> As for our setting, if we similarly impose a noise term in the generative model of $M_i$, we would also need to add a diagonal structure in our loss objective $\ell(X)$ to offset the bias due to the norm of the noise. The magnitude of this diagonal term can also be calculated based on the noise variance.
>
> Notice, however, that our treatment of the diagonal structure allows the loss objective to remain the same between the theoretical analysis and the practical implementations across Sections 3 and 5. We believe this treatment is appropriate when the variance of the (white) noise is relatively small, which is the case in the datasets that we experiment with. In addition, this also unifies the loss objective in our setting, which simplifies the presentation.
>
> We have now added a Remark to discuss this related work and clarify this extension at the end of Section 3.
>
> **Comment 2:** “It would be helpful to provide a precise and explicit definition of the "one-side matrix completion" problem in both the abstract and the introduction. In addition, it would be helpful to add some discussion about why this problem is important and challenging.”
>
> **Response to Comment 2:** Thanks for the constructive feedback. We have now revised the abstract and also the introduction to clarify the problem setup of one-sided matrix completion. In particular, we included the suggested description that we assume that $M$ is a tall matrix, and the goal is to recover the row space of $M$.
>
> We have now also added descriptions in the introduction to better motivate this setting. In particular, in the datasets we experiment with, the averaged $n$ over $d$ is around $10$.
>
> Lastly, we have revised the introduction to highlight two technical challenges to recover the row space $T$ in the ultra-sparse sampling regime. In particular, the first involves dealing with the non-random missing patterns of $hat T$, while the second involves analysing the optimization landscape of $\ell(X)$.
>
> **Comment 3:** “In Theorem 3.4, it would be helpful to discuss the assumption that the number of users n must be much larger than the number of items d, and explain why this assumption is reasonable in real-world applications.”
>
> **Response to Comment 3:** Thanks for the suggestion. We have added a paragraph after presenting Theorem 3.4 to discuss the validity of this assumption. In particular, we now connect this assumption more explicitly with the dataset examples evaluated in our experiments.
>
> **Comment 4:** “The Hájek estimator is known to exhibit finite-sample bias in general. Since the paper shows it is unbiased in this setting, it would be helpful to highlight and explain this result more clearly in the introduction.”
>
> **Response to Comment 4:** Thanks for this excellent suggestion. We have now revised the introduction to highlight this observation more clearly. In particular, this unbiasedness property in the finite sample regime is one of the key motivating observations for our theoretical result.
>
> **Comment 5:** “The first extension, recovering individual entries of M, is closely aligned with the standard matrix completion problem. Therefore, it would be helpful to expand this extension a bit and discuss the practical scenarios where this extension is feasible.”
>
> **Response to Comment 5:** We have revised this section to expand on this extension, in particular, clarifying that when each row contains a sufficiently large number of entries, then we could also achieve accurate user-level recovery based on the recovered row space of $M$.
>
> **Comment 6:** “The second extension, concerning differential privacy, is currently a bit hard to follow and would benefit from revision for clarity.”
>
> **Response to Comment 6:** Thanks for this suggestion. We have revised this section to improve the clarity of this extension. In particular, we now better clarify the Gaussian mechanism, and we elaborate on the sensitivity-based analysis for the differential privacy of the Gaussian mechanisms.

---

> ### Author Response · Authors · 2025-08-21
> **Response to Reviewer 8mat (continued)**
>
> **Response to minor issues and typos:** Thanks very much for the detailed reading! We have revised the paper to address these typos and misspellings and added clarifications when needed.

---

### Review · Reviewer_SM4K · 2025-11-08

**Summary Of Contributions:**

The paper studies the recovery of the row space of a matrix under an ultra-sparse sampling regime. It first analyzes the variance between the Hájek estimator and the Horvitz–Thompson estimator over the observed entries. Then, the author proposes a gradient descent algorithm for the unobserved entries and investigates its sample complexity. Finally, the paper provides empirical validation on both synthetic and real-world datasets to demonstrate the effectiveness of the proposed method.

**Audience:**

Yes

**Broader Impact Concerns:**

No concerns towards ethical implications.

**Claims And Evidence:**

Yes

**Requested Changes:**

Typo

1. In the first line of Section 2, "With loss of generality" should be corrected to "Without loss of generality."

2. In the first line after Assumption 3.3 "Assuming that $µ(U)$ is a fixed value that does not row with d". What does this mean?

Question:

1. In Equation (4), the first term is $M^4_{k, i}$ whereas in Equation (6) it is $M^2_{k, i}$. Why the paper states "By comparing equation (4) to equation (6), we see a reduction in the second term from equation (4), which is at the same order as the leading term" ?

**Strengths And Weaknesses:**

Strengths

1. The paper provides a strong theoretical analysis of matrix row-space completion under an ultra-sparse sampling regime. The analysis includes a variance comparison between the Hájek and Horvitz–Thompson estimators over the observed entries, as well as a study of the sample complexity of the gradient descent algorithm applied to the unobserved entries.

2. The proposed method, grounded in solid theoretical foundations, demonstrates significant performance improvements on both synthetic and real-world datasets.

3. The incorporation of incoherence regularization effectively reduces recovery error, highlighting the practical value of the theoretically motivated penalty term.

Weakness

1. The paper assumes that the rank $r$ is known for both the theoretical analysis and empirical evaluation. However, in real-world applications, the true rank is typically unknown, which may limit the practical applicability of the proposed approach.

---

> ### Author Response · Authors · 2025-11-09
> **Response to Reviewer SM4K**
>
> We thank the reviewer for carefully reading our manuscript and providing constructive feedback.
>
> **Response to Weakness 1:** Thanks for this comment. We agree that in practice, the true rank is typically unknown and thus, one would first need to determine the rank of the target matrix. To address this issue in our approach, we have added an ablation analysis in Appendix C.3, where we vary the rank of the target matrix and compare different ranks via cross-validation splits. The analysis shows that this cross-validation analysis helps select a rank parameter that gives strong empirical results.
>
> On the theory side, it is an interesting question to estimate the true rank from the observed data. This can be achieved by running SVD on the empirical second-moment matrix and carefully analyzing its spectrum induced by the noise variables. We have added a remark in Section 4 to discuss this extension and leave the full theoretical analysis to future work. We have also added a paragraph at the end of the introduction to clearly state this limitation as part of our sample complexity bound.
>
> **Response to Typos:** Thanks for pointing out both issues. We have addressed them in the revised manuscript. In addition, we carefully revised the paper throughout to proofread typos and improve exposition.
>
> **Response to Question 1:** Thanks again for your careful reading. In Equation (6), it should also be $M_{k, i}^4$. Then, these two equations will have the same order, and the variance reduction comes from the second term on the right-hand side of Equation (4), since this term is always negative. We have fixed this issue in the revised manuscript as well.
>
> Please let us know if you have any further questions or comments during the discussion stage.
>
> Thank you.

---

### Decision · Action_Editor_Ukrf · 2025-12-29

**Recommendation:** Accept as is

**Audience:**

Yes

**Audience Explanation:**

Yes. The paper addresses the practically important and challenging problem of matrix completion under an ultra-sparse regime, which is highly relevant for large-scale recommendation systems. The proposed Hajek estimator and Hajek-GD method are novel and potentially useful in practice. The theory and algorithm are likely to attract TMLR readers interested in recommendation systems, large-scale machine learning, and optimization.

**Claims And Evidence:**

Yes

**Claims Explanation:**

The paper proposes a new method Hajek-GD for ultra-sparse matrix completion. The theoretical analysis justifies the advantage (lower variance) of newly proposed Hájek estimator compared to existing HT estimator.  The empirical experiments show that Hajek GD significantly outperforms exsiting matrix completion baselines in a bunch of real world datasets. Overall, the effectiveness of Hajek-GD is supported by enough evidence.